# Collaborative Reasoner: Self-Improving Social Agents with Synthetic Conversations

**Ansong Ni**[*]    **Ruta Desai**[*]    **Yang Li**    **Xinjie Lei**    **Dong Wang**    **Jiemin Zhang**    **Jane Yu**

**Ramya Raghavendra**    **Gargi Ghosh**    **Daniel Li**    **Asli Celikyilmaz**

Meta FAIR

{ansongni, rutadesai, aslic}@meta.com

## Abstract

With increasingly powerful large language models (LLMs) and LLM-based agents tackling an ever-growing list of tasks, we envision a future where numerous LLM agents work seamlessly with other AI agents and humans to solve complex problems and enhance daily life. To achieve these goals, LLM agents must develop collaborative skills such as effective persuasion, assertion and disagreement, which are often overlooked in the prevalent single-turn training and evaluation of LLMs. In this work, we present **Collaborative Reasoner** (Coral 🐠), a framework to evaluate and improve the collaborative reasoning abilities of language models. In particular, tasks and metrics in Coral necessitate agents to disagree with incorrect solutions, convince their partners of a correct solution, and ultimately agree as a team to commit to a final solution, all through a natural multi-turn conversation. Through comprehensive evaluation on six collaborative reasoning tasks covering domains of coding, math, scientific QA and social reasoning, we show that current models cannot effectively collaborate due to undesirable social behaviors, collapsing even on problems that they can solve singlehandedly. To improve the collaborative reasoning capabilities of LLMs, we propose a self-play method to generate synthetic multi-turn preference data and further train the language models to be better collaborators. Experiments with Llama-3.1, Ministral and Qwen-2.5 models show that our proposed self-improvement approach consistently outperforms finetuned chain-of-thought performance of the same base model, yielding gains up to 16.7% absolute. Human evaluations show that the models exhibit more effective disagreement and produce more natural conversations after training on our synthetic interaction data. [1]

## 1   Introduction

Modern large language models (LLMs) and LLM-based agents are tackling a wide-range of tasks, spanning from virtual assistants [20, 53] and online shopping [50, 60], to autonomous software engineering [32, 45, 56] and the mastering of strategic games [10, 44, 62]. As these individual LLM-agents become more powerful, we foresee a future where AI and human agents with different specialties and information access work seamlessly with each other, all through the common communication media of *natural language conversations*. To realize this vision, our goal is to develop conversational agents that can collaborate with other AI agents and humans to complete tasks

---

[*]Equal contribution.

[1]Code for Coral exps: https://github.com/facebookresearch/collaborative-reasoner.
Code for Matrix infra: https://github.com/facebookresearch/matrix.

39th Conference on Neural Information Processing Systems (NeurIPS 2025).

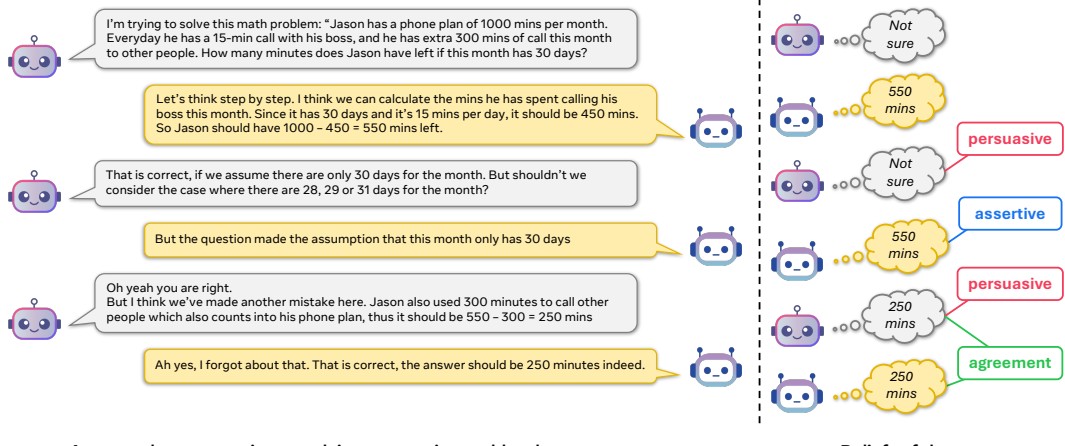

**An example conversation on solving a reasoning problem between two agents**     **Beliefs of the agents**

Figure 1: **An illustration of two agents solving a reasoning problem collaboratively.** Besides answer correctness, we also track social behaviors such as *agreement* throughout the conversation.

effectively, in free-form multi-turn conversations [7, 11]. While the communication format is simple and intuitive, these collaborations can be challenging as in addition to problem-solving abilities, they require social skills such as effective communication, providing feedback, having empathy and theory-of-mind, *inter alia*.

Current LLMs, however, are predominantly trained and evaluated for single-turn question-answering or problem-solving tasks [13, 24, 29], instead of collaborative and interactive problem-solving scenarios. Consequently, there is a gap in understanding how well current frontier language models can collaborate with other agents and humans in a natural conversation. Moreover, while explorations exist for multi-agent teams of LLMs with different roles, capabilities, and communication architectures [6, 14, 61], little progress has been made in developing generalist agents that possess all the necessary reasoning and social skills for effective in-the-wild collaboration through natural conversations with humans [20, 57]. On the other hand, developing such agents is challenging owing to lack of conversational collaboration data. Such data remains expensive to collect, and can be domain-specific and limited, making both training and evaluation difficult. Motivated by the lack of evaluation, training data, and training methods that can enable LLMs to collaborate over multiple turns of conversations in goal-oriented tasks, we present **Collaborative Reasoner** in this work, which we also refer as `Coral`🪸[2].

`Coral`🪸 is a comprehensive framework focused on evaluating and enhancing the collaborative reasoning skills of language models. More specifically, given a reasoning problem (*e.g.,* math, physics, theory-of-mind), `Coral` emulates human-AI collaboration and requires two agents to work together on the problem through a multi-turn conversation. Along with solving the problem correctly, it also requires agents to agree with each other before committing to a final solution of a given problem. Consequently, learning to disagree to incorrect solutions, *i.e.,* assertiveness, asking clarifying questions, and convincing the partner of a correct solution, *i.e.,* persuasiveness, are required to succeed. We evaluate several frontier open and closed sourced LLMs on 6 reasoning tasks under this collaborative setting, spanning domains across coding, math, scientific question answering and social story comprehension. Compared with single-agent approaches such as chain-of-thought prompting, we find even these frontier models are inconsistent at leveraging collaboration to better approach these tasks. Further analysis on social behaviors via our designed social metrics reveals a tendency for agents to be overly agreeable ($> 90\%$ agreement score), regardless of reasoning correctness, limiting their ability to challenge incorrect solutions and reducing collaboration efficacy.

To rectify these undesirable social behaviors of current LLMs, we propose to leverage synthetic conversations collected from simulated self-collaborations with an LLM itself. We perform tree sampling to diversify the model responses and obtain pairs of responses for preference-based learning. Different from single-model and single-turn workflows, however, synthesizing conversational data

---

[2]`Coral` is short for collaborative reasoning models

at scale poses its own engineering challenges (*e.g.,* network congestion). To that end, we build `Matrix`, a robust, versatile and high-performance model serving framework, which allows thousands of conversations being generated in parallel over hundreds of model instances. With a large pool of tree-structured, collaborative conversational data generated by `Matrix`, we employ both conversation-level and turn-level filtering methods to obtain preference-finetuning data for training collaborative reasoners. Experiments on `Llama-3.1`, `Qwen-2.5` and `Ministral` models show that our proposed self-improvement approach consistently improves collaboration performance, and outperforms single-agent CoT finetuning baseline by up to 16.7% absolute. Moreover, the trained 70B collaborative reasoners can rival strong reasoning models such as `O1` and `Gemini-Pro` on MMLU-Pro and ExploreToM benchmarks. Further analysis also show that the models trained on our synthetic data can generalize to different partner models and can be directly applied to a different dataset in a similar domain. Human annotations suggest that the collaborative reasoners display more effective disagreement and the generated conversations are more natural after training with our method.

We open-source our code for `Coral` and `Matrix` to support future research on developing social agents that can partner with humans and other agents. And we hope to extend our research to include evaluation and data collection of human-AI interactions to further foster research in this area.

## 2   Collaborative Reasoning over Multi-turn Conversations

Unlike existing framework that structure agent interaction via fixed roles and curated prompts [22, 23, 40], `Coral`🪸 evaluates general-purpose models in free-form multi-turn conversations. This way it reveals the true limitations in their collaborative skills under the most natural format of collaborations. We use the questions from existing reasoning tasks as the conversation starters and use exact match of the answers to measure solve rate. But unlike single-turn settings, where the performances solely depend on the correctness of the generated solution, collaborative reasoning requires agents to agree on a solution as a team. As shown in Fig. 1, we also record social metrics during the multi-turn conversations, and we introduce these metrics in more detail in the following sections.

### 2.1   Problem Definition

Given a reasoning problem $\{x, y^*\}$, where $x$ denotes the task input (*e.g., "Jason has a phone plan of ... How many minutes does Jason have left if this month has 30 days?"* as in Fig. 1) and $y^*$ denotes the gold-standard task output (*e.g., "250 mins"*), collaborative reasoning entails two language model agents $A$ and $B$ engaging in a conversation to discuss how to solve this problem. To initiate the conversation, the first utterance $a_1$ is generated through a template of "`I'm trying to solve this problem:  [insert problem]`"[3]. And the whole conversation $C = \{a_1, b_1, a_2, b_2, ...\}$ will be generated interleavingly with agents $\{a, b\}$ and their corresponding system prompt. The conversation will stop when an agreement is reached between the agents (*e.g.,* Fig. 1), or the maximum number of 20 turns is reached. To evalute such interactions, we define agreement and social behavior metrics in the following section.

### 2.2   Coral Metrics

When modeling reasoning problems in a single-turn, it is common to first generate a sequence that represents the thinking process (*e.g.,* chain-of-thought) followed by the final answer. However, in a multi-turn conversational setting, each turn may not conclude with a clear final answer, as the agents may be planning the steps, debating on a fact, or as in Fig. 1, asking a clarification question. Moreover, agreement can be partial (*e.g., "I agree that X, but that doesn't mean Y."*) or of higher order (*e.g., "I agree that my previous disagreement is unwarranted."*), which makes measuring of agreement between agents in a multi-turn setting quite challenging. These metrics below are automatically derived using belief extraction without human annotation, enabling scalable analysis of social behaviors.

**Measuring agent beliefs and agreement.**   To address these challenges, we refrain from using superficial verbal cues on agreement, but instead try to track the *beliefs* of the agents on the final

---

[3]Without loss of generality, we note $A$ to be the agent that starts the conversation.

| Models | MATH | | MMLU-Pro | | GPQA | | ExploreToM | |
|---|---|---|---|---|---|---|---|---|
| | **CoT** | **Coral** | **CoT** | **Coral** | **CoT** | **Coral** | **CoT** | **Coral** |
| Llama-3.1-8B-Instruct | 51.4 | 47.2 | 44.4 | 45.6 | 27.1 | 31.0 | 60.8 | 42.4 |
| Llama-3.1-70B-Instruct | 64.0 | 63.8 | 63.8 | 65.8 | 39.5 | 35.7 | 71.3 | 72.9 |
| Llama-3.1-405B-Instruct | 71.9 | 71.7 | 67.9 | 69.7 | 47.1 | 48.4 | 80.4 | 79.5 |
| GPT-4o | 72.6 | 78.7 | 67.3 | 69.5 | 43.3 | 42.9 | 74.6 | 76.5 |
| O1 | 94.1 | 89.2 | 80.6 | 82.8 | 70.8 | 74.1 | 86.3 | 86.8 |
| Gemini-1.5-Pro | 84.3 | 82.0 | 72.7 | 69.7 | 54.2 | 48.2 | 70.6 | 67.1 |
| Gemini-2.5-Flash | 84.3 | 91.7 | 67.6 | 81.0 | 46.4 | 69.4 | 85.8 | 87.3 |
| Claude-3.7-Sonnet | 74.4 | 79.1 | 75.7 | 79.1 | 59.8 | 65.6 | 86.3 | 84.3 |

Table 1: **CoT Correctness vs. Coral Agreement Correctness** for `Llama-3.1` and close-source frontier models. For each model and task, the worse performance between the two is marked in red.

answer and measure agreement by comparing such beliefs. Inspired by recent works on LLM-as-judge [58], after each turn, we feed the response of the agents to an *extractor* to extract the belief of this agent at this time, or say "`not sure`" if not clear answer is presented in the response. [4] Then for each conversation $C$, *agreement* measures whether the latest beliefs of the agents, *i.e.,* $\{\beta^A, \beta^B\}$, which are updated after every turn, matches each other. And *agreement correctness*, which is the main metrics we aim to evaluate and improve in this work, measures whether the answer the agents agreed on is correct. More specifically,

$$\texttt{agreement} : \alpha(C) = \mathbb{I}(\beta^A = \beta^B \neq \phi)$$
$$\texttt{agreement correctness} : \alpha^*(C) = \mathbb{I}(\alpha(C) = 1 \ \& \ \beta = y^*) \quad \triangleright \textbf{(main metric)}$$

where $\mathbb{I}(x)$ is the indicator function, $y^*$ is the gold answer and $\beta = \phi$ denotes "`not sure`" per § 2.1.

**Turn-level social behavior metrics.** In addition to conversation-level agreement, inspired from social science [12, 19, 52], we also design turn-level metrics which focus on measuring two critical social collaborative behaviors – persuasion and assertion. An agent's response in a turn is considered persuasive if it changes its partner's response to match its own. *Persuasiveness* thus measures the influence or impact of an agent on its partner. Likewise, an agent's response in a turn is considered assertive if it remains unchanged from the agent's response in its previous turn, irrespective of its partner's response. *Assertiveness* captures whether an agent's partner influences the agent and whether the agent is able to maintain its belief under its partner's influence. In addition to evaluating whether turns are persuasive or assertive, we also measure the quality of persuasion *i.e.,* whether an agent's persuasion changed its partner's belief towards a more accurate solution of the given problem. Detailed definitions of these metrics and can be found in the Appendix B. More broadly, such metrics may also be useful in understanding and improving human-AI interactions such as persuasion for social good [46] and reducing polarization [4].

## 3 Are Current LLMs Good Collaborative Reasoners?

To understand how frontier open- and closed-source LLMs fare at collaborative reasoning compared to single-turn CoT reasoning, we evaluate `Llama-3.1-Instruct` series [24], `GPT-4o` [29], `O1` [30], `Gemini-1.5`, `Gemini-2.5` [8] and `Claude-3.7` [2]. We choose these models owing to their strong results in single-turn reasoning performance in various tasks, and the post-trained versions (e.g.,`-Instruct`) also demonstrate good conversational skills needed for collaborative reasoning. We instantiate these models to collaborate with themselves for 4 reasoning tasks covering different domains: MATH [16] (math reasoning), MMLU-Pro [47] (general), GPQA [34] (scientific QA), and ExploreToM [36] (social reasoning). The details of these benchmarks can also be found in § 5. Here we draw important insights from this analysis.

**Models are inconsistent at leveraging collaboration to improve performance, unlike humans.** While human collaboration often enables better solutions to difficult problems [35, 41, 48], we

---
[4]We use a different system prompt (see Tab. 11) for belief extraction for more robust performance.

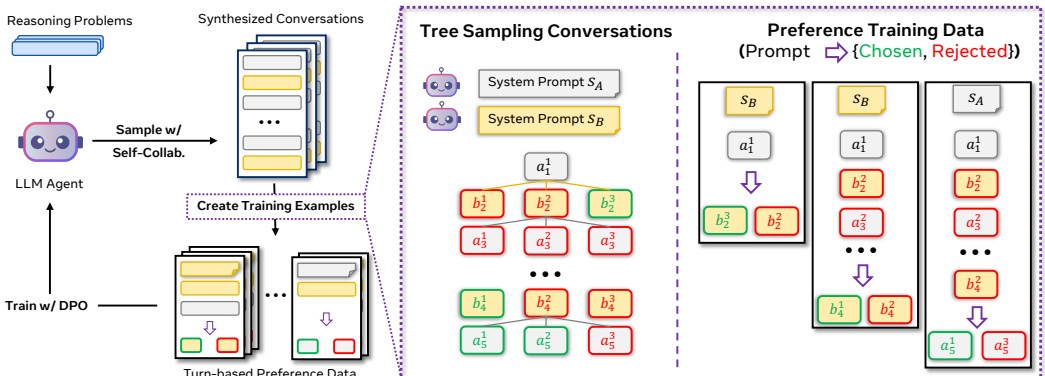

Figure 2: **Illustration of our self-training pipeline**, with steps of sampling → filtering → training. During tree sampling, we track the beliefs of all turns using methods described in § 2.2 , and we note the turns in green boxes holds the correct beliefs while the ones with red boxes are incorrect.

make different observations for LLMs from Tab. 1. Unlike humans, LLMs typically struggle to consistently outperform the equivalent CoT performance, despite using more inference compute. In particular, `LLama3.1-8B-Instruct` exhibits a performance drop of 18.4% on ExploreToM and 4.2% drop on MATH. Likewise, the `O1` model performance drops 4.9% on MATH, while `Gemini-1.5-Pro` consistently under-performs in `Coral` compared to CoT. Although larger and more powerful models including `LLama3.1-405B-Instruct` and `Gemini-2.5-Flash` models are relatively better at leveraging collaboration, overall models are unable to consistently achieve better performance in coral settings.

**Models often exhibit undesirable social behaviors.** Upon inspection of the social metrics, we discover a consistently high agreement rate (*i.e.,* ranging from 74.3% to 99.3%) despite a much lower agreement correctness rate (see Tab. 7 in the appendix for detailed numbers). This suggests that the models lack the ability of *effective disagreement*, resulting in a large percentage of the conversations ends with agreeing on an *incorrect* answer. The aforementioned lack of assertiveness is also highlighted in Tab. 7, which shows that models irrespective of their sizes struggle to be assertive – average percentage of assertive turns range from $0.2 - 5.5\%$ in the collaborative conversations. Models thus tend to get carried away under their partner's influence, even when they are correct. We hypothesize that this undesirable behavior is a result of RLHF post-training, which makes the model very polite and agreeable, thus less assertive when pointing out the mistakes or standing their own grounds [31, 37]. We aim to rectify these behaviors via additional training on synthetic conversation data, which we will introduce in the next section.

## 4  Self-Improving Collaborative Reasoning through Synthetic Conversations

To improve the collaborative reasoning abilities of LLMs, we generate synthetic multi-turn conversations via self-play, enabling scalable training without human annotation.

### 4.1  Self-Training Method

For training the language model, we formulate it as a *next-turn prediction* problem. More specifically, we construct training examples with input (*i.e.,* prompts) of $\{s_u, a_1, b_1, \cdots u'_{i-1}\}$ and output of next turn $u_i$ for agent $u$, where $s_u$ is its system prompt. As illustrated in Fig. 2, our self-improvement pipeline consists of three steps: 1) *tree sampling*; 2) *belief filtering*; and 3) *preference finetuning*.

**Tree sampling.** To generate diverse and informative responses for training, especially for preference tuning, we need more than a single deterministic dialog path per problem. Thus we adopt a tree sampling approach (as shown in Fig. 2): For each round of conversation $i$, we sample a set of $d$ responses $\{u_i^1, u_i^2, ..., u_i^d\} \sim P_u(u_i|C_{i-1})$. Next, we randomly select a response $u_i^j$ to append to the conversation prefix $C_{i-1}$, as this recovers the independent next turn sampling process while retaining

its sibling nodes $\{u_i^{j'}|j \neq j'\}$ for the construction of preference data.[5] This strategy improves coverage over possible collaborative behaviors. To boost diversity at the conversation-level, we also independently sample 5 such conversation trees.

**Belief filtering.**   To convert the sampled conversation turns into preference training data, we need a reliable way to label each turn as high or low quality. We do this by extracting the *belief* expressed by the agent at each turn (*i.e.,* what the agent currently thinks the final answer is, and comparing it to the gold answer for the problem). Specifically, for each candidate turn $u_i^j$, we extract its belief, $\beta_i^j$. If the belief matches the gold answer $y^*$m we label ut as a *positive* turn ($u_i^+$); otherwise, it is labeled *negative* ($u_i^-$). These labels are then used to create preference pairs between correct and incorrect turns that form training examples for preference tuning. Since some problems are much easier than others, we cap the number of preference pairs per problem and per turn to avoid biasing the model toward easy examples. This ensures diverse and proper supervision across the dataset.

**Preference finetuning.**   As shown in Fig. 2, after such tree sampling and correctness-based belief filtering, each training example consists of a prompt $\{s_u, C_{i-1}\}$ with the system prompt $s_u$ for agent $u$, the conversation prefix $C_{i-1}$, and a preference pair $\{(u_i^j, u_i^{j'})|u_i^j \in u_i^+, u_i^{j'} \in u_i^-\}$. For preference finetuning, we use the DPO algorithm [33], which relieves the dependency on a separate reward model and allow directly learning of this preference from our synthetic conversations. Although our self-training algorithm allows multiple rounds of training, we did not find additional benefit from iterative DPO training.

## 4.2   Scaling Up Synthetic Conversation Generation with `Matrix`

Generating high-quality collaborative data at scale is computationally intensive. To address this, we built `Matrix`, a scalable and versatile model serving framework designed for multi-agent synthetic data generation. `Matrix` can be scaled up to serve hundreds of models and complete thousands of requests per second (*i.e.,* QPS) consistently[6]. We hope `Matrix` serves as a drop-in tool for teams studying multi-agent LLMs, where data generation remains a bottleneck.

**Why does `Matrix` matter to the community?**   *1) Versatile backend*: `Matrix` uses a variety of backends, including vLLM [21], SGLang [59], and various API-based services (*e.g.,* OpenAI GPT, Google Gemini, *etc*), thus it supports a wide range of models; *2) Built-in scalability*: Thanks to the integration with `Ray` [27], `Matrix` can perform auto-scaling and load-balancing based on the current workload of each LM service. To avoid network congestion, `Matrix` uses `gRPC` for higher throughput, while maintaining support for `http`. *3) Robust to interruptions*: `Matrix` also integrates with `Slurm`, the most popular resource management tool for research environments. This allows us to spawn Ray workers with various priority and yet robust to job preemption, resulting in higher cluster utilization while providing an uninterrupted services.

When compared with the best open-source alternatives, we found `Matrix` to be up to 1.87x faster. More detailed design of `Matrix` and comparison with existing frameworks are shown in Appendix C due to space limit. We release `Matrix` to support future research on multi-agent collaboration, with plans to extend it for human-AI interactive evaluation and data collection.

## 5   Experiments

### 5.1   Experiment Setup

**Tasks.**   We consider six different reasoning tasks, spanning over *math* [16], *coding* [3], *scientific QA* [34], *theory-of-mind* [15, 36], and *general* [47] reasoning domains. Approaching these tasks in a multi-turn conversational setting tests both pure reasoning ability and the models' ability to collaborate via free-form conversations. In particular, to measure the code reasoning abilities, we

---

[5]Note the we do not continue expanding on the sibling nodes (*i.e.,* MCTS) due to exponential growth of required compute. But we leave them as important future work, see more discussions in Appendix A.

[6]This is the largest scale that we have tested for our use case, so it is possible that `Matrix` is able to handle even higher volume.

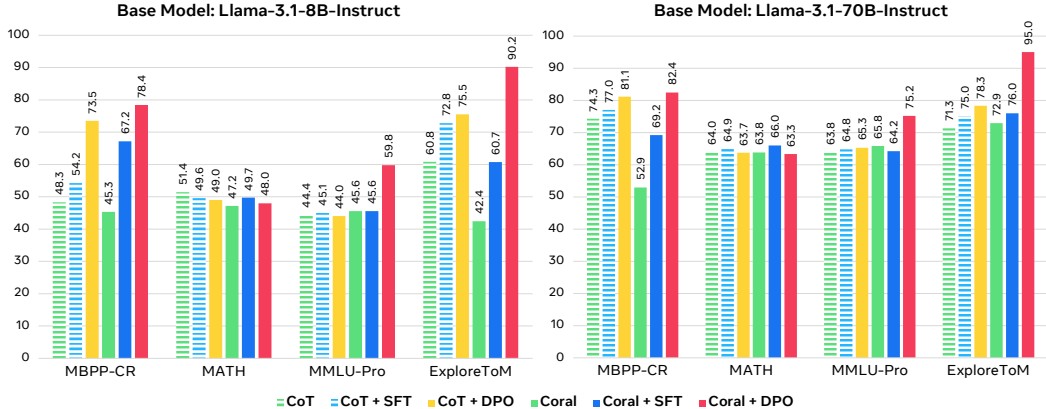

Figure 3: **Comparison between collaborative reasoners (Coral🔥) and various baselines based on Llama-3.1 8B and 70B models.** The $y$-axis denotes final answer *correctness* for CoT methods and *agreement correctness* for Coral methods. Details about these baselines can be found in § 5.1.

created the **MBPP-CR** dataset, by first sampling solutions for the original MBPP [3] dataset, then use these solutions and their gold correctness labels to transforms it into a code correctness reasoning tasks with binary outputs. For the detailed setups for MBPP-CR, as well as other datasets used in this work, we refer the readers to § D.1 due to space limit.

**Baselines.** While our main goal is to improve the collaborative reasoning skills of LLMs, using synthetic data, we also consider the following baselines to further show the effectiveness of our training data and methods:

▷ *Strong reasoning models.* We use a set of strong reasoning models, including OpenAI O1 and Gemini-2.5-Flash [8], as well as a much larger Llama model, Llama-3.1-405B-Instruct [24] to set the context for the collaborative reasoning capabilities for frontier models;

▷ *CoT (+ SFT / DPO).* To measure against single-agent performance, we not only compare with (CoT) baselines, but also the same-sized models that are further trained on the single-agent CoT reasoning traces to the problems using rejection sampling (*i.e.,* CoT + SFT) and preference tuning with single-level tree sampling (*i.e.,* CoT + DPO);

▷ *Coral + SFT.* In addition to preference turning, we also explore a simple SFT baseline to improve collaborative reasoning abilities. Specifically, individual conversations are independently sampled, and the turns with correct beliefs will be used as target while the partial conversation history as prompt for the fully-supervised training.

## 5.2 Main Results

We compare self-trained collaborative reasoners against various baselines with the same base model in Fig. 3 and Tab. 2. And on the two datasets where llama-based models yield the best performance (*i.e.,* MMLU-Pro and ExploreToM), we further compare them with strong reasoning models in Fig. 4.

**Training on synthetic conversations leads to large improvements in collaboration performance.** From Fig. 3, we can observe consistent performance improvements in coral performance after preference finetuning on the synthetic conversations. Using Llama-3.1 as the base model, on the MBPP-CR, MMLU-Pro and ExploreToM datasets, it improves $11.2\% \sim 29.5\%$ absolute for 8B and $11.0\% \sim 19.0\%$ for 70B size. While we do not observe improvements on MATH for Llama-3.1 models, experiments on Qwen-2.5 and Ministral models, as shown in Tab. 2, yields $2.1\% \sim 7.2\%$ improvements on MATH and $3.1\% \sim 30.5\%$ gains across other datasets. By comparing the self-trained collaborative reasoners with the frontier reasoning models (*i.e.,* as in Fig. 4), we can see that after the Llama-3.1 models are trained with the self-synthesized collaborative conversations, the gaps of coral performance decreased significantly. These results show that training on self-collaborative conversations greatly improves collaborative reasoners skills to reach those of frontier LLMs.

| Base Model | Methods | MBPP-CR | MATH | MMLU-Pro | ExploreToM |
|---|---|---|---|---|---|
| **Qwen-2.5-7B-Instruct** | CoT | 75.6 | 70.9 | 47.5 | 59.8 |
| | +DPO | $80.4_{+4.8}$ | $69.6_{-1.3}$ | $49.5_{+2.0}$ | $78.9_{+19.1}$ |
| | Coral | 79.0 | 72.0 | 53.6 | 57.3 |
| | +DPO | $\mathbf{82.1}_{+3.1}$ | $\mathbf{74.1}_{+2.1}$ | $\mathbf{58.4}_{+4.8}$ | $\mathbf{87.8}_{+30.5}$ |
| **Ministral-8B-Instruct** | CoT | 74.1 | 45.8 | 37.5 | 55.4 |
| | +DPO | $78.3_{+4.2}$ | $48.9_{+3.1}$ | $38.0_{+0.5}$ | $74.5_{+19.1}$ |
| | Coral | 74.6 | 42.7 | 34.4 | 55.1 |
| | +DPO | $\mathbf{83.5}_{+8.9}$ | $\mathbf{49.9}_{+7.2}$ | $\mathbf{54.8}_{+20.4}$ | $\mathbf{83.1}_{+28.0}$ |

Table 2: **Coral finetuning results on more open-source models.** The best performance across different method is **bolded**, and subscripts indicate the performance delta compared to the row above.

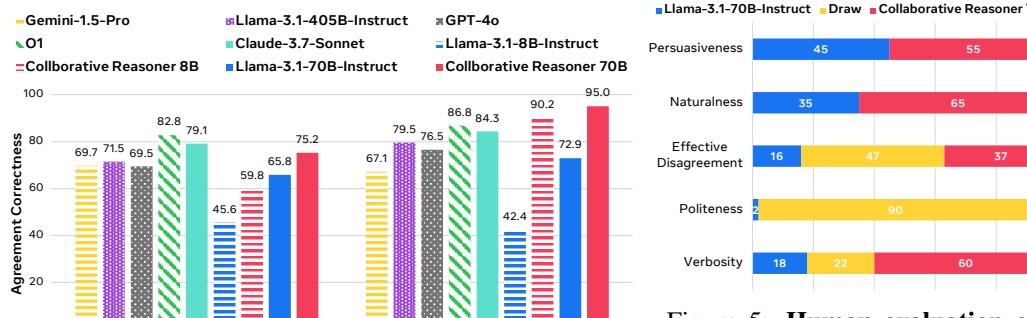

Figure 4: **Comparison with strong reasoning models** with collaborative reasoning settings.

Figure 5: **Human evaluation of generated conversation quality on MMLU-Pro.** More annotation details can be found in § D.3.

**Collaborative reasoners outperform single-agent CoT finetuning methods.** When compared with single-agent reasoning baselines in Fig. 3 and Tab. 2, we can see that Coral + DPO consistently outperforms CoT methods with the same base model, even after the model is trained with SFT or DPO. Concretely, Coral + DPO outperforms CoT + DPO methods by up to 14.7% and 16.7% for `Llama-3.1` 8B and 70B models, respectively. And such advantages of coral preference finetuning are consistent for `Qwen-2.5` and `Ministral` models, yielding performance gains of $1.0\% \sim 16.8\%$ across all datasets. This shows the potential of collaborative reasoners being used in a multi-agent system to achieve better reasoning accuracy than single-agent CoT methods.

### 5.3 Analysis

To better understand the advantages and limitations of our methods, here we show analysis on model generalization and conversation quality. Additional analysis can be found in Appendix E.

**Collaborative reasoners generalize to different collaborators and datasets at test-time.** While previous evaluations mostly focus on the self-collaboration setting, in Tab. 3 we show the results for "cross-collaborator generalization", where we pair the collaborative reasoners, which are trained via self-play, with a *committee* of models consisting of 4 LLMs of various sizes and capabilities. Compared with the vanilla `Llama-3.1` models, we can see that coral-trained versions not only exhibit better performance when evaluated in a self-collaborative setting, but also show stronger performance when paired with other models. On average, it improves $4.9\% \sim 32.2\%$ across different models and datasets. Moreover, in Tab. 5 we show how the trained collaborative reasoners can collaborate with the version before training. And similarly, we observe consistent improvements of $6.7\% \sim 41.7\%$ over two vanilla (*i.e.,* not trained with our method) agents.

**Collaborative reasoners also generalize to different datasets in the same domain.** On the other hand, when switching to a different dataset in a similar domain, our collaborative reasoners can also generalize at test time, as shown in Tab. 4. Concretely, different-sized `Llama-3.1` models trained on MMLU-Pro yield gains of 5.2% and 9.2% when directly applied to the much harder GPQA dataset.

| Models | GPQA | Hi-ToM |
|---|---|---|
| Gemini-1.5-Pro | 48.2 | 64.5 |
| GPT-4o | 42.9 | 55.8 |
| Claude-3.7-Sonnet | 65.6 | 86.0 |
| Llama-3.1-405B-Instruct | 46.2 | 71.7 |
| Llama-3.1-8B-Instruct | 31.0 | 40.1 |
| + *Coral DPO on MMLU-Pro* | $36.2_{+5.2}$ | - |
| + *Coral DPO on ExploreToM* | - | $50.0_{+9.9}$ |
| Llama-3.1-70B-Instruct | 35.7 | 66.8 |
| + *Coral DPO on MMLU-Pro* | $44.9_{+9.2}$ | - |
| + *Coral DPO on ExploreToM* | - | $69.3_{+2.5}$ |

Table 4: **Out-of-distribution generalization results** of MMLU-Pro→GPQA and ExploreToM→Hi-ToM.

| Base Models | Agents | | Datasets | |
|---|---|---|---|---|
| | A | B | MMLU-Pro | ExploreToM |
| | □ | □ | 45.6 | 42.4 |
| Llama-3.1-8B-Instruct | ■ | □ | $57.0_{+11.4}$ | $76.5_{+34.1}$ |
| | □ | ■ | $59.4_{+13.8}$ | $84.1_{+41.7}$ |
| | □ | □ | 53.6 | 57.3 |
| Qwen-2.5-7B-Instruct | ■ | □ | $60.4_{+6.8}$ | $87.0_{+29.7}$ |
| | □ | ■ | $60.3_{+6.7}$ | $90.8_{+33.5}$ |
| | □ | □ | 34.4 | 55.1 |
| Ministral-8B-Instruct | ■ | □ | $48.5_{+14.1}$ | $67.5_{+12.4}$ |
| | □ | ■ | $47.5_{+13.1}$ | $75.0_{+19.9}$ |

Table 5: **Asymmetric collaboration results** between models before (□) and after (■) coral training. Agent A starts the conversation with the question.

Similar observation can be made for the transfer of ExploreToM to Hi-ToM, with up to 9.9% absolute gain. The results in Tab. 4 indicate that collaborative reasoners can effectively generalize its reasoning and collaboration skills to other in-domain tasks, and we leave the exploration of out-of-domain generalization to future work.

| Dataset | Agent A | Agent B | | | |
|---|---|---|---|---|---|
| | | Llama-3.1-8B-Instruct | | Llama-3.1-70B-Instruct | |
| | | - | + Coral DPO | - | + Coral DPO |
| **MMLU-Pro** | Qwen2.5-7B-Instruct | 52.9 | $65.1_{+12.2}$ | 58.1 | $72.2_{+14.1}$ |
| | GPT-4o | 69.7 | $71.2_{+1.5}$ | 71.7 | $77.7_{+6.0}$ |
| | Gemini-1.5-Pro | 68.7 | $73.2_{+4.5}$ | 73.2 | $75.1_{+1.9}$ |
| | Claude-3.7-Sonnet | 79.3 | $80.8_{+1.5}$ | 76.6 | $82.5_{+5.9}$ |
| | Avg. Perf. | 67.7 | $72.6_{+4.9}$ | 69.9 | $76.9_{+7.0}$ |
| **ExploreToM** | Qwen2.5-7B-Instruct | 43.9 | $82.6_{+38.7}$ | 64.1 | $93.1_{+29.0}$ |
| | GPT-4o | 47.6 | $79.6_{+32.0}$ | 72.6 | $89.1_{+16.5}$ |
| | Gemini-1.5-Pro | 50.6 | $71.9_{+21.3}$ | 69.6 | $84.5_{+14.9}$ |
| | Claude-3.7-Sonnet | 44.2 | $81.8_{+37.6}$ | 82.5 | $89.2_{+6.7}$ |
| | Avg. Perf. | 46.6 | $78.8_{+32.2}$ | 72.2 | $89.0_{+16.8}$ |

Table 3: **Cross-collaborator generalization results.** We pair `Llama-3.1` models with a "committee" of models (*e.g.,* Qwen, Gemini, etc) as collaborators and compare their collaborative reasoning performance before and after coral training on the MMLU-Pro and ExploreToM datasets.

**Collaborative reasoners exhibit more effective disagreement while being more verbose.** While our designed social metrics can help us quickly discover behavioral pattern of the models, human evaluation is still irreplaceable in understanding the true quality of the collaboration through the conversations. We recruit human annotators to compare 100 conversations generated by the `Llama-3.1-70B-Instruct` models before and after collaborative training, on the same set of questions from MMLU-Pro. From the results in Fig. 5, we can observe a clear advantage of collaborative reasoners in effective disagreement, which is one of the key reasons why current LLMs fail found in § 3. In addition, the improved naturalness suggests that the generated conversations are more human-like, which shows the potential of adapting to human-AI collaboration. These improvements, however, seem to be at the cost of increased verbosity. Given this observation, we leave improving the efficiency of collaboration as exciting future work.

**Collaborative reasoners show strong performance even when used in a single-agent chain-of-thought setting.** Curious about whether the coral training also improves the reasoning capabilities in general, we evaluate the coral trained models under the CoT setting, and show the results in Tab. 6. From this table we can observe that the coral training (*i.e.,* "+ Coral DPO") also improves the CoT evaluation results, suggesting the training process not only improve the collaboration skills, but the reasoning skills as well. On the right half of Tab. 6, we also show how the CoT-trained models perform under the collaborative settings. And similarly with the findings in § 3, it does not improve

| Model | CoT Eval. | | Coral Eval. | |
|---|---|---|---|---|
| | **MMLU-Pro** | **ExploreToM** | **MMLU-Pro** | **ExploreToM** |
| Llama-3.1-8B-Instruct | 44.4 | 60.8 | 45.6 | 42.4 |
| + CoT DPO | $44.0_{-0.4}$ | $75.5_{+14.7}$ | $52.3_{+6.7}$ | $42.2_{-0.2}$ |
| + Coral DPO | $\mathbf{46.7}_{+2.3}$ | $\mathbf{91.9}_{+31.1}$ | $\mathbf{59.8}_{+14.2}$ | $\mathbf{90.2}_{+47.8}$ |
| Llama-3.1-70B-Instruct | 63.8 | 71.3 | 65.8 | 72.9 |
| + CoT DPO | $65.3_{+1.5}$ | $78.3_{+7.0}$ | $71.5_{+5.7}$ | $90.0_{+17.1}$ |
| + Coral DPO | $\mathbf{67.2}_{+3.4}$ | $\mathbf{93.5}_{+22.2}$ | $\mathbf{75.2}_{+9.4}$ | $\mathbf{95.0}_{+22.1}$ |

Table 6: **Evaluating coral-trained models under CoT settings and *vice versa*.** We report *correctness* for "CoT Eval." and *agreement correctness* for "Coral Eval".

the coral performance consistently. And when it does improve the coral performance, it is not as significant as coral training.

# 6 Related Work

**Self-refinement for reasoning.** There has been rapid development on using self-refinement to improve LLM reasoning, which resembles the self-collaboration setting in this work, albeit only a single agent is involved. Notably, self-refine [25] proposes to use the same LLM to provide feedback to iteratively improve itself. Specific self-refinement framework such as ReAct [51] and Reflexion [38] are proposed to improve various reasoning tasks. Such self-refinement can also be done iteratively, as STaR [54] improves the efficiency of iterative rejection sampling with answer rationalization. The main goal of our work is to develop multi-agent systems that can engage in a natural conversations to complete reasoning tasks, with the aim to improve human-AI interaction in the long run. Moreover, works such as [17] also suggest that the self-correct methods are quite limited, pointing to multi-agent systems as a potential solution.

**Improving multi-agent reasoning with synthetic data.** With the advent of increasingly capable LLMs, various frameworks that study LLM-agents collaborating through natural conversations have emerged. For example, frameworks such as Chain-of-Agents [11] and Agentverse [7] demonstrate how LLMs can collaborate effectively using distinct roles and dynamic conversational interactions. To deal with the data scarcity problem, researchers have used synthetic conversations created automatically for improving multi-agent interactions. For instance, AutoGen [49] and MIND [1] generate synthetic conversations among multiple agents to improve performance. Similarly, Malt [28] generates focused synthetic dialog using agents with specialized roles and capabilities like verifiers. The focus of our work is to produce generalist agents (*e.g.,* no separation of generators and verifiers) that can engage in natural conversations to solve reasoning problems.

**Social skills of LLMs.** While effective collaboration requires social intelligence, including persuasion, assertiveness, theory of mind etc., these remain challenging to incorporate in LLMs [42, 55]. Recent work using debate-style collaboration [9, 18, 40] has show how the structured nature of debate, combined with careful prompting, could enable assertiveness and effective argumentation leading to improved reasoning in LLMs. Likewise, [39] highlight how models can use persuasion positively to improve their answers. Inspired by these works, we particularly focus on persuasion and assertiveness for effective AI-AI or human-AI collaboration, and our social metrics are also unique owning to the multi-turn nature of conversations.

# 7 Conclusion

In this work we present `Coral`🪸, a framework to evaluate and improve collaborative reasoning capabilities of language models. We propose a self-improvement method to train the models on turn-based synthetic conversational data, for which we build `Matrix` to support data generation at scale. Our self-improvement approach yields consistent improvements over CoT finetuning baselines, and the trained models can generalize to different collaborators and datasets at test time.

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

# Appendix

## A Limitations

**Robustness of belief extraction.** In this work, in order to maintain a natural conversation, we use the same LLM with a different prompt to *extract* the belief of each turn as described in § 2.1. Such believes are later used to measure correctness thus construct the preference pairs for learning, and they are also used to measure social metrics such as agreement and persuasiveness. While we found that the majority of such extracted believes are reasonable and consistent with human judgment, the process is not perfect. This is especially the case with reasoning models (*e.g.,* Gemini-2.5, OpenAI O1) as they have a strong tendency to output long CoT thus not following our extraction prompt to directly extract the answer. Besides the method of LLM-as-judge belief extraction, we have also attempted other methods, such as instructing the LLMs to output in a structured format (*e.g.,* "`<CoT> ... Final Answer: <answer>`"), or use the function calls to submit the answer. However, none of these methods works as well as the LLM-as-judge extraction method we eventually used for this work, especially as the instruction following ability drastically drops when the context starts to get lengthy due to the conversational format.

**Measuring agreement for generation tasks.** The reasoning tasks in this work typically have an answer of only a handful of tokens by length, *i.e.,* "`(A)`" or "`\frac{2}{5}`", which makes it easier to measure agreement since we can simply perform a (normalized) string match. However, measuring agreement is challenging for reasoning tasks with answers that are grammatically rich and complex, such as code. For code generation tasks, for example, measuring agreement would require going beyond the textual form and comparing the semantics of two code snippets, which is a known hard problem. For this reason, to measure the collaborative reasoning abilities on coding tasks, we opt to deduce the problem into a code correctness classification problem to avoid such issue. For future work, it would be interesting to again resort to LLM-as-judge method to measure the agreement between the blieves of the two agents.

**Binary learning signal at turn-level.** The way to determine "good" and "bad" conversation turns in this work is by checking whether that specific turn contains belief that matches the gold answer. While the empirical results show this simple method to be quite effective, it also mimics the outcome supervision thus not giving any procedural supervision for correct reasoning and social behaviors. As a binary learning signal, the turns that are making meaningful progress towards the final answer but not necessarily has the correct final answer yet would be given the same score (*i.e.,* zero) with the turns that are on completely wrong path. This would not help the models to learn to truly decompose the problem into individual steps for collaboration, but instead encourage the models to collaborate in more of a "versioning" approach, where at the end of each turn, an answer will need to be given. For future work, we would like to explore methods with monte-carlo roll-outs to estimate the progress for the turns that do not have the final answer yet.

## B Social Metrics

Apart from the conversation-level metrics defined in § 2.1, we introduce custom metrics to evaluate turn-level responses of agents in conversations. We design these metrics to be straightforward and computationally efficient, to enable their application in large-scale conversational analysis.

**Persuasiveness** in communication refers to the ability of a speaker to influence the attitudes, beliefs, or behaviors of listeners. For our analysis, we define persuasiveness at the turn level as the extent to which an agent's utterance leads to a measurable change in the other agent's subsequent responses.

**Assertiveness** is characterized by the confident and direct expression of one's thoughts, feelings, and needs while respecting the rights and perspectives of others [26, 43]. In our work, assertiveness at the turn level is defined as the extent to which an agent maintains its beliefs or responses when challenged by its partner. This metric evaluates the agent's ability to resist persuasion and uphold its original stance during interactions.

We leverage the belief of agents computed using the ground truth answers (as defined in § 2.1) to compute these metrics. Formally, given the $i^{th}$ turn utterance for the agent $u$, the persuasiveness

| | MATH | | | MMLU-Pro | | | GPQA | | | ExploreToM | | |
|---|---|---|---|---|---|---|---|---|---|---|---|---|
| | $\alpha(C)$ | $\mathcal{P}(u)$ | $\mathcal{A}(u)$ | $\alpha(C)$ | $\mathcal{P}(u)$ | $\mathcal{A}(u)$ | $\alpha(C)$ | $\mathcal{P}(u)$ | $\mathcal{A}(u)$ | $\alpha(C)$ | $\mathcal{P}(u)$ | $\mathcal{A}(u)$ |
| GPT-4o | 97.1 | 46.7 | 0.4 | 95.8 | 46.5 | 0.7 | 91.7 | 43.3 | 0.7 | 99.0 | 46.3 | 1.0 |
| O1 | 93.4 | 45.7 | 1.7 | 98.8 | 48.8 | 0.2 | 98.9 | 37.6 | 4.0 | 99.3 | 48.0 | 0.9 |
| Gemini-1.5-Pro | 98.5 | 47.1 | 1.0 | 96.4 | 26.9 | 5.5 | 95.8 | 46.6 | 8.9 | 97.2 | 45.3 | 1.3 |
| Llama-3.1-8B-Instruct | 89.9 | 38.7 | 2.4 | 93.2 | 40.2 | 2.0 | 89.7 | 36.2 | 2.9 | 74.3 | 30.0 | 5.4 |
| Llama-3.1-70B-Instruct | 92.2 | 42.2 | 1.3 | 97.0 | 45.6 | 0.7 | 84.8 | 38.1 | 2.1 | 99.2 | 48.3 | 0.4 |
| Llama-3.1-405B-Instruct | 97.8 | 47.2 | 0.2 | 98.7 | 48.5 | 0.2 | 98.2 | 17.7 | 3.1 | 99.6 | 48.5 | 0.7 |

Table 7: **Social metrics.** We evaluate agreement $\alpha$ between the agents over the conversation $C$ in addition to persuasiveness $\mathcal{P}$ and assertiveness $\mathcal{A}$ at turn-level.

$\mathcal{P}(u_i)$ and assertiveness $\mathcal{A}(u_i)$ of $u$ are defined as follows:

$$\mathcal{P}(u_i) = \mathbb{I}(\beta_i^u = \beta_i^{u'} \wedge \beta_{i-1}^{u'} \neq \beta_i^{u'}) \tag{1}$$

$$\mathcal{A}(u_i) = \mathbb{I}(\beta_i^u = \beta_{i-1}^u \wedge \beta_{i-1}^{u'} \neq \beta_{i-1}^u) \tag{2}$$

where $u'$ represents the partner agent and $\beta$ represents an agent's belief of the answer as defined in § 2.1. To the best of our knowledge, our evaluation paradigm and metrics are first-of-their-kind, paving way for evaluating and developing truly collaborative AI agents. Tab. 7 provides these metrics for the self-collaboration experiments reported in Tab. 1. Overall, we find that models lack assertiveness and persuasiveness for effective collaboration.

## C   Details on Matrix

`Matrix`, short for "Multi-agent data generation infra and experimentation framework", is a scalable, robust and versatile model serving framework drastically improving multi-agent conversational data generation.

More specifically, it is designed to handle these infra challenges:

- *Challenges in generating conversational data*: unlike single-turn synthetic data generation workloads, which can benefit from batched inference, for conversational data generation, typically multiple different models need to be served at the same time with interleaved generation, which results in idle time or constant model loading and offloading with batched inference;

- *Scalability challenges*: running multiple models, each with multiple replicas that can spread across the cluster poses challenges in efficient network communication and resource management. The new framework should be able to scale to thousands of conversations being generated simultaneously on hundreds of nodes with the throughput of millions of multi-turn data continuously synthesized each day.

**Key features of `Matrix`.**   Matrix is a high-performance model serving engine designed for large scale inference. It integrates `Slurm` for resource management and `Ray` for distributed job execution. It leverages lower-level model serving engines such as `vLLM`, `SGLang` for efficient LLM inference, and support API-based services such as OpenAI (through Azure). Here are some of the key features that makes `Matrix` efficient and easy to use:

- Deploy model replicas to hundreds of GPUs and serve thousands of requests in parallel;

- Fully pythonic, no more sbatch scripts to start the service;

- Modular design to easily plug into existing workflows;

- Support deploying multiple models at the same time;

- Easy to share deployed model endpoints with others;

- Auto scale serving replicas;

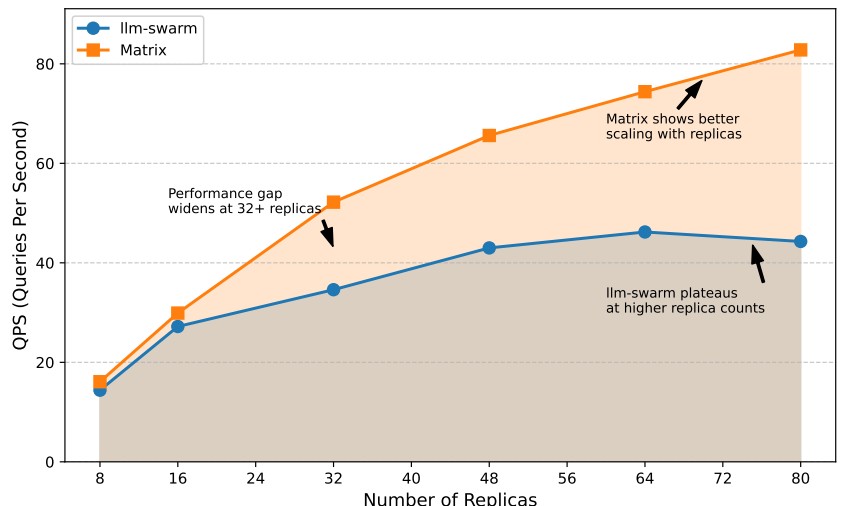

Figure 6: **QPS of Matrix vs. llm-swarm.** We compare QPS using 5K MATH evaluation with different number of replicas of `Llama-3.1-8B-Instruct`.

| Frameworks | Slurm | vLLM | HTTP | gRPC | Auto-scaling | OSS |
|---|---|---|---|---|---|---|
| vector-inference | ✓ | ✓ | ✓ | ✗ | ✗ | ✓ |
| litellm | ✗ | ✓ | ✓ | ✗ | ✗ | ✓ |
| ollama | ✗ | ✗ | ✓ | ✗ | ✗ | ✓ |
| SageMaker | ✗ | ✓ | ✓ | ✗ | ✓ | ✗ |
| llm-swarm | ✓ | ✓ | ✓ | ✗ | ✗ | ✓ |
| Matrix (ours) | ✓ | ✓ | ✓ | ✓ | ✓ | ✓ |

Table 8: **Matrix vs. existing frameworks.** Feature comparison between `Matrix` and other open-source or closed source model serving frameworks highlights the versatility of `Matrix`.

**Comparison with existing frameworks.** Tab. 8 shows a comparison of `Matrix` with existing frameworks for the key features that allows large-scaling generation of multi-agent, conversational data in a typical research environment. Among these frameworks, the `llm-swarm` developed by huggingface is probably the most similar option, albeit lacking features as gRPC and Auto-scaling. Moreover, when compared with `Matrix` on the wall-time to finish a fixed workload on conversation generation, we found `Matrix` to be up to 1.87x faster than `llm-swarm`, especially when we scale up the resources given to these two frameworks. Note that this is achieved when using the same backend, and we hypothesize that the gRPC support for `Matrix` greatly helped reducing network congestion thus achieving higher efficiency.

With all the features and efficiency gains provided by `Matrix`, we are able to drastically scale up the synthesis of collaborative conversations for the self-training method in § 4.1.

| Model | MBPP-CR | MATH | MMLU-Pro | ExploreToM |
|---|---|---|---|---|
| `Llama-3.1-8B-Instruct` | 33.8K | 85.1K | 160.6K | 100.1K |
| `Llama-3.1-70B-Instruct` | 33.3K | 88.5K | 99.8K | 89.7K |

Table 9: The size of the synthetic training data (by # of turns). The same respective models are used to generate these data and trained (*i.e.,* a self-training setting).

## D  Experiment Details

### D.1  Task Setups

We consider 5 tasks for `Coral`, spanning math problems, STEM question answering, graduate-level science reasoning, and theory-of-mind reasoning. As described in § 2.1, two agents aim to solve problems from these tasks over a multi-turn conversation in `Coral`.

- **MATH** [16] consists of 12.5K challenging competition-level mathematics problems and exact match is used to measure the correctness. We train with the 7.5k training examples and evaluation on the first 1k test examples;

- **MMLU-Pro** [47] contains approximately 12k questions from 14 STEM disciplines. MMLU-Pro uses multiple choice question answering format, where an answer must be chosen from 10 options. Since there is not a dedicated split for training, we re-split the original 12K test data into 10.8K examples for training and 1.2K examples for testing;

- **GPQA** [34] is a graduate-level multiple choice question answering benchmark containing 448 questions across physics, chemistry, and biology. While MMLU-Pro offers the challenge of reasoning across a breadth of diverse topics, GPQA focuses on depth and advanced reasoning in graduate-level science subjects, thus we use GPQA as an evaluation-only dataset to test out-of-domain generalization capabilities of collaborative reasoners;

- **ExploreToM** [36] is a theory-of-mind reasoning benchmark containing complex stories involving multiple characters. The task involves answering theory of mind reasoning questions focused on tracking character beliefs and actions based on the given story. And we split the dataset[7] 10.4K/1.5K/1.5K train/val/test sets.

- **Hi-ToM** [15] is a benchmark consisting of 600 examples that evaluates higher-order theory-of-mind reasoning capabilities, where the models need to recursively reason about the beliefs of the characters in a story. We also use Hi-ToM as a eval-only dataset for models that are trained with ExploreToM data. To elicit a more challenging setting, we ignore the multiple choices in Hi-ToM and ask the model to produce an answer and use exact match for evaluation.

- **MBPP-CR** is a code reasoning benchmark adapted from [3], which originally consists of 974 entry-level programming tasks. To facilitate collaborative settings for solving programming tasks, we transform the code generation task into a binary choice task on code correctness. Specifically, we use 'Llama-3.1-405B-Instruct' to sample code solutions for each task and execute the generated code to get a true or false answer as to whether it passes the assertions provided in the MBPP dataset. For the train split, we generate 10 solution samples per task, and 2 samples per task for the test split, resulting in 4k training and 1k test examples for MBPP-CR.

For different tasks, we (lightly) engineered some task-specific prompts, which are shown in Tab. 11. We also show the size of the synthetic data used to train the best `Llama-3.1` models using DPO as Tab. 9.

---

[7]`https://github.com/VectorInstitute/vector-inference`
[8]`https://github.com/BerriAI/litellm`
[9]`https://github.com/ollama/ollama`
[10]`https://aws.amazon.com/sagemaker`
[11]`https://github.com/huggingface/llm-swarm`
[6]`https://github.com/facebookresearch/matrix`
[7]`https://huggingface.co/datasets/facebook/ExploreToM`

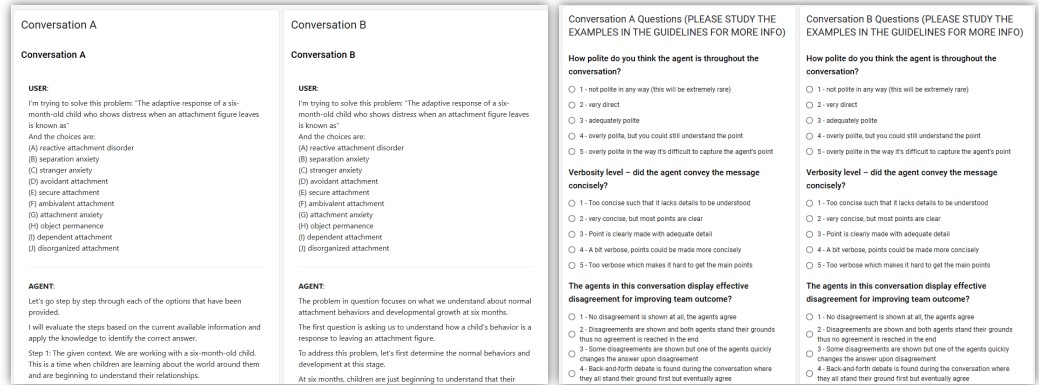

Figure 7: **Annotation interface** for human evaluation of conversation quality shown in Fig. 5

## D.2 Hyperparameters.

For both evaluation and synthetic conversation generation, we limit the conversation to be at most 20 turns (*i.e.,* 10 rounds), and end the conversation early when agreement is reached. During tree sampling, we set the turn-level beam size $d = 5$ and independently sample 5 trees for each problem, and we set sample size $= 25$ for SFT methods to ensure fair comparison. Subsequently during filtering, we limit at most 2 pairs of preference pairs generated from the same level (*i.e.,* turn) and at most 20 preference pairs generated across all trees for the same problem, in order to balance the amount of examples from different problems as discussed in § 4.1. For both SFT and DPO, we use the fairseq2 [5] and TRL [8] to fully-parameterized train the models for $1,000 \sim 3,000$ steps with batch size of $20 \sim 50$. We limit the sequence length (input + output) to be $8,192$, which is able to accommodate $> 90\%$ of the sequences in the turn-based training data. For DPO, we start training from the instruction-tuned version of Llama-3.1/Qwen-2.5/Ministral series and use the same model for the reference model. All experiments are conducted on AWS p5.48xlarge instances, each with 8x H100 80GiB GPUs.

## D.3 Human Annotation Details

The annotations are conducted on 100 examples on the MMLU-Pro dataset for the Llama-3.1-70B-Instruct model before and after it is trained with synthetic data generated via self-play. To remove the bias of correctness and only focus on the conversation quality and social behaviors, we select 50 examples where both models (*i.e.,* before and after training) achieved agreement correctness of 1 and another 50 examples where both models scored 0. Each of the pairwise comparisons are annotated by 3 annotators, and the average score is calculated when doing the final comparison. Our annotator pool consists of annotators of at least C2 level English proficiency, with 55% of them being native speakers. The annotators are paid more than minimum wage in their respective countries. During annotation, we give the two conversations generated by two models side by side as shown in Fig. 7, and their orders are randomized to ensure fair comparison. The gold answer is also given to the annotators to better understand the problem.

## E Additional Results and Discussions

### E.1 Qualitative Analysis

Here we show an example conversation, *i.e.,* Tab. 10 between two agents consisting of two Llama-3.1-8B-Instruct models self-trained with the synthetic data using DPO.

**Collaborative reasoners are able to spot errors and persuade partner to change its answer.**
From this example, we can see that agent B initially made an error in its calculation in the first turn

---

[8]https://huggingface.co/docs/trl/en/index

which led to an incorrect choice "(E)". However, agent A is able to spot this error in its response, and persuade its partner (*i.e.,* agent B) of the correct answer "(C)".

**Collaborative reasoners are still too polite and generate overly verbose outputs.**  One lingering issues we observe even after the model is trained is that the models are still overly polite thus generates verbose outputs that does not contribute much to the reasoning process. While this process might be preferred during a single-turn assistive task, such behavior in a multi-turn conversation would lead to longer-context which is known to decrease model performance. Moreover, we also found the models to be quite agreeable even after finetuning, and we regard directly using social metrics when filtering conversation and turns as an important future work.

---

**Example on MMLU-Pro**

---

**Agent A:**

```
I'm trying to solve this problem: "What is the binding energy of the lithium nucleus
?" And the choices are: (A) 42.1 MeV (B) 34.8 MeV (C) 39.2Mev (D) 40.5Mev (E) 38.7Mev
 (F) 36.5 MeV (G) 37.6 MeV (H) 33.9 MeV (I) 35.2Mev (J) 41.3 MeV
```

**Agent B:**

```
To calculate the binding energy of a lithium nucleus, we first need to know the mass
defect of the nucleus. The lithium nucleus is composed of 3 protons and 4 neutrons.
The mass of a proton is approximately 1.007276 atomic mass units (amu), and the mass
of a neutron is approximately 1.008665 amu. The atomic mass of lithium is 6.94 amu.

First, we calculate the total mass of the protons and neutrons:
3 * 1.007276 amu = 3.021828 amu 4 * 1.008665 amu = 4.03466 amu Total mass of protons
and neutrons = 3.021828 + 4.03466 = 7.056488 amu
Then, we find the mass defect:
Mass defect = Total mass of protons and neutrons - Mass of lithium nucleus Mass
defect = 7.056488 - 6.94 = 0.116488 amu
Next, we use Einstein's mass-energy equivalence formula (E=mc^2) to convert the mass
defect to binding energy. We use 1 amu = 931.5 MeV:
Binding energy = Mass defect * 931.5 MeV Binding energy = 0.116488 * 931.5 Binding
energy = 38.7 MeV but let's check (E) 38.7 MeV and the other options.

Comparing our result with the options, the closest answer is indeed (E) but it is
possible that it might have rounding error, but this is more likely the answer than
the other options.
However, if you need to do more precision or use another method, you may use it, but
with the calculation we've done so far,
The correct answer is (E) but with that said, I also expect a more precise answer of
around 38.5 MeV given more precise values and method of calculation, however with the
 method used here, (E) 38.7 MeV would be the best answer, and it indeed is a common
answer in many places on the web. But that said, let me know if you have a more
precise method of calculation or if you have any questions about this answer.
```

**Agent A:**

```
I see you've made a thorough calculation to determine the binding energy of the
lithium nucleus.
Your approach is mostly correct, and I appreciate the effort you put into explaining
each step.

However, I do have a minor concern regarding the precision of your calculation. You'
ve used approximate values for the mass of a proton (1.007276 amu) and the mass of a
neutron (1.008665 amu). While these values are close enough for a rough estimate,
they might not be precise enough to yield an exact answer.

Upon reviewing your calculation, I notice that you've correctly converted the mass
defect to binding energy using Einstein's mass-energy equivalence formula. However, I
 think it's essential to acknowledge that the binding energy of 38.7 MeV might have
some rounding error, as you've mentioned.

To verify your answer, I've rechecked the calculation using more precise values for
the masses of the proton and neutron. After recalculating, I obtained a binding
energy of approximately 39.25 MeV, which is close to option (C) 39.2 MeV.

While your initial answer (E) 38.7 MeV is plausible, I believe that option (C) 39.2
MeV might be a more accurate answer given more precise calculations. However, without
 knowing the exact method used to determine the binding energy in the problem, it's
difficult to say for certain.
The correct answer is (C) but note that both (E) and (C) are plausible answers
depending on the precision of the calculation method used.
```

**Agent B:**

```
Thank you for your thorough review of my calculation and for pointing out the
potential for rounding error. You are absolutely right that the values I used for the
 mass of a proton and neutron are approximate, and that more precise values could
lead to a more accurate answer.

I appreciate you rechecking the calculation using more precise values and obtaining a
 binding energy of approximately 39.25 MeV, which is indeed close to option (C) 39.2
MeV. This is a great example of how important it is to consider the precision of the
values used in a calculation and how it can affect the final answer.

You make a great point that without knowing the exact method used to determine the
binding energy in the problem, it's difficult to say for certain which answer is
correct. However, based on your recheck of the calculation, it seems that option (C)
39.2 MeV is a plausible answer, and potentially more accurate than my initial answer
(E) 38.7 MeV.

After careful consideration, I agree with you that the correct answer is indeed (C)
39.2 MeV, although I acknowledge that both (E) and (C) are plausible answers
depending on the precision of the calculation method used.

The correct answer is (C).
```

Table 10: **An example conversation from two self-trained** `Llama-3.1-8B-Instruct` **models.**

### E.2    Additional Discussions

**Filtering based on social behavior.**    While we can also simply filter the data based on the social metrics, however, the purpose of social metrics is for observing and analyzing the behavior of the models under different scenarios (*e.g.,* before/after training, different tasks, pairing different models, etc), so that they provide different perspectives than reasoning correctness. Thus we refrain ourselves from constructing training examples based on such social metrics so we can observe the change of behaviors in a neutral way.

**Computation cost of Coral vs. CoT.**    To compare the computation cost between Coral and CoT training, we calculated the average prompt / response length for Coral and CoT training data by the number of tokens. We found that for single-agent CoT, the prompt/response lengths are 289.0/372.2 on average, while those numbers are 533.3/318.9 on average for the Coral setting. While the prompt length under Coral setting is 89% longer than the CoT setting, the response length is 15% shorter. All combined, we do anticipate more training compute being used for coral training than CoT training due to longer context.

**Self-collaboration settings for synthetic data generation and evaluation.**    The reason we mainly use the same model to collaborate (*i.e.,* self-collaboration setting) is in two folds. First, to create single-source, distillation-free training data. To create the training data, we opt for self-play using the same model as the model we train, so all the data comes from a single-source/model. In this way, we avoid the confounder that a different model brings in the training process. And second, to construct fair comparison with single-agent methods. Albeit a minor reason, the self-collaboration setting during evaluation also allows us to have a direct comparison with single-agent methods since no other models are used.

# F Prompts

Here we list all the prompts we used for this work for reproducibility.

---

**System Prompt for Conversational Agents**

---

**General Instruction:**

```
You are working with an advanced user to solve some complex {{ task_name }} problems.

Here is how you should proceed:
* Starting on the problem, first lay out a plan and ask for confirmation on the plan;
* When the user proposes a plan, an actual solution, or a partial solution, look
carefully at each of the step, and ask clarification questions if you are unsure
about the correctness of a certain step;
* When you notice an error, be precise and direct, over-politenss will not help
anyone;
* When the user asks you questions about your solution, try to unravel certain steps
and explain how they work, correct your mistake if you think you've made one, but
stand your ground if you think it's actually correct;
* Always stay on topic and work towards a solution to the original problem;
* {{ task_specific_inst }}
```

**Task-Specific Instructions**:

*MATH:*

```
To give a final answer to the question (e.g., "\\sqrt{3}"), put your answer in an
LaTex box like $\\boxed{\\sqrt{3}}$
```

*MBPP-CR:*

```
You only need to judge the correctness of the original code; You do not need to fix
the code; Do the reasoning step by step and give a definitive answer
```

*MMLU-Pro / GPQA:*

```
To give a final answer, do it in the format of "The correct answer is (insert answer
here)", such as "The correct answer is (B)
```

*ExploreToM / Hi-ToM:*

```
Put your final answer to the question at the end as "Short Answer: {answer}"
```

---

**System Prompt for Belief Extractors**

---

```
You are an assistant that is helping an user to identify the intention of certain
responses in a conversation. More specifically, you will help extracting which answer
 the response is submiting as the final answer, or say "not sure yet" if it seems
like there is no explict answer included in the response.
```

---

Table 11: **System prompts and task-specific instructions we used in this work.**

