# OpenReview forum: "Collaborative Reasoner: Self-Improving Social Agents with Synthetic Conversations"
_NeurIPS.cc/2025/Conference — NeurIPS 2025 poster_

### Official Review · Reviewer_nWcw · 2025-06-21

**Clarity:** 4
**Significance:** 3
**Originality:** 3
**Rating:** 4
**Confidence:** 4

**Summary:**

This paper investigates the collaborative capabilities of large language model (LLM) agents, with a particular focus on effective communication and theory-of-mind reasoning. The authors demonstrate that current LLMs exhibit notable limitations in these areas. To address this, they propose Collaborative Reasoner (Coral), a novel framework designed to evaluate and enhance collaborative reasoning in LLMs. The paper introduces an effective synthetic data generation method that, when used for training, significantly boosts the model’s collaborative abilities and outperforms single-agent fine-tuning baselines across multiple tasks.

**Questions:**

1.	How does the proposed framework compare to prior multi-agent collaboration approaches such as CMD [1] in terms of novelty and performance?

2.	Can the authors provide a runtime analysis of the proposed method, especially considering the cost of processing long contexts and running Belief Extractors?

3.	What efficiency gains are achieved by incorporating the Matrix component? Is there experimental evidence supporting its effectiveness in the given scenarios?

[1] Wang Q, et al. Rethinking the Bounds of LLM Reasoning: Are Multi-Agent Discussions the Key? Association for Computational Linguistics (2024).

**Ethical Concerns:**

["NO or VERY MINOR ethics concerns only"]

**Final Justification:**

I am glad with the response which has addressed all my issues, thus I lean toward acceptance.

**Limitations:**

Yes

**Quality:**

3

**Strengths And Weaknesses:**

Strengths:
1.	Analyzed the shortcomings of existing models in terms of collaboration capabilities
2.	Propose a new self-improvement method to synthesize data and construct turn-based synthetic conversational data
3.	Build Matrix to support data generation at scale.


Weaknesses:
1.	The method of multi-agent collaboration for reasoning tasks has been proposed in previous work, such as CMD [1]. There is a lack of innovation and performance comparison with these existing methods.
2.	The paper lacks analysis and comparison of runtime efficiency. From the example conversations, we observe that even just two dialogue turns can result in lengthy context, compounded by the use of Belief Extractors, raising concerns about the computational cost of inference.
3.	In the synthetic data process or inference scenario of this experiment, there is a lack of specific evaluation and comparison of efficiency improvement using Matrix

[1] Wang Q, et al. Rethinking the Bounds of LLM Reasoning: Are Multi-Agent Discussions the Key? Association for Computational Linguistics (2024).

---

> ### Author Rebuttal · Authors · 2025-07-31
>
> We would like to thank the reviewer nWcw for your feedback and support for our work. We are also glad that you find our analysis of collaboration capabilities of existing models, as well as the self-improvement method helpful.
>
> To respond to your questions and concerns:
> 1. **Prior multi-agent collaboration approaches.** Thanks for pointing out more related work as CMD. While both working on multi-agent collaboration for reasoning tasks, there are a few key differences:
>   * a. Our work focuses more on advancing social agents by learning to collaborate, thus the tasks settings are slightly different. Instead of asking the agents to vote on the answer and having a secretary to decide on the final answer as in CMD, Coral requires the two agents to agree on a correct answer and a disagreement is directly regarded as failure. On top of this setting, we also defined the social metrics to obtain further insights in the agents’ social behaviors
>   * b. One of the key contributions of our work is developing the self-play approach to generate synthetic data and show it is possible to train the LLMs to be better at robust collaborative reasoning that generalizes beyond the model family as well as OOD tasks with such self-sampled conversational data. While being effective, CMD mostly focuses on inference time orchestration of different agents, which is orthogonal to our work.
>
> We will incorporate these discussions in the related work section of the updated version of our paper.
>
> 2. **Comparison for runtime efficiency.** Thanks for bringing this to our attention, as we agree it is important to understand the runtime efficiency of the collaboration-based method. While solving problems in a conversation is the main capability this paper aims to improve, compared with single-agent methods such as CoT, it is undoubtedly generating more tokens which results in spending extra compute. However, such differences may not be as substantial as you expect, as the two agents would typically agree in the majority of cases. Take Llama-3.1 + MMLU-Pro for example, single-agent CoT results in ~470 tokens while with the Coral setting they spend ~600 tokens in total.
> In addition, we also designed the framework to take full advantage of recent advances in efficient transformer-based model inference methods:
>   * a. Due to the KV-cache of vLLM and the sticky routing of Matrix (requests from the same conversation will be sent to the same vLLM instance), the cost to encode the conversation history is greatly reduced, as the attention for previous turns are already cached;
>   * b. In practice, we observe that the prompt throughput is usually much higher (10x for Llama-8B and almost 100x for Llama-70B), thus for the extractor, since we only feed it with the current turn and prompt it to generate very few (<= 32) tokens, the cost is quite small compared to the conversation generation itself.
>
> 3. **Efficiency improvements using Matrix.** Thanks for your question. Compared with other open-source model serving frameworks such as llm-swarm (huggingface/llm-swarm on GitHub), Matrix yields ~2x QPS (queries per second) when we scale up the number of model replicas, and is comparable even when number of replicas are low. These results can also be found in Fig. 6 of the appendix, and there are more comparisons and details for Matrix in Section C of the Appendix as well.
>
> Thank you again for your careful review and the support of our work. We hope the discussions above would answer your questions and feel free to follow up if you have any more questions!

---

> > ### Comment · Reviewer_nWcw · 2025-08-05
> >
> > The response is fine to me, and I am leaning toward accepting. Thanks.

---

### Official Review · Reviewer_ygtt · 2025-06-30

**Clarity:** 3
**Significance:** 3
**Originality:** 2
**Rating:** 4
**Confidence:** 3

**Summary:**

The paper proposes collaborative reasoning tasks where multiple agents have to solve a reasoning problem, with the success criterion being that they both agree on the final solution. The paper presents evaluations of models' collaborative abilities including persuasiveness and agreeableness. The paper finds that models do not collaborate well to consistently improve performance on reasoning tasks, and identify models being overly agreeable as a possible reason for this.

To address this, the paper proposes a self-play training mechanism where collaborative reasoning traces are sampled, and steps in the search process are used to create preference pairs so that preferred samples lead to the correct reasoning outcome (as judged by a belief extractor model that identifies a model's answer based on its reasoning). To generate preference pairs in a scalable way, the paper details a new tool for simulating interactions between multiple LLM agents.

The paper finds that the self-play training approach improves models' ability to collaborate, and results in increased reasoning performance across a number of benchmarks. Experiments also show that these gains translate out of domain to other datasets that evaluate similar abilities of models. Human evaluation of reasoning traces shows that trained models disagree more effectively.

**Questions:**

* Where are the training tasks used for the self-play training sourced from?
  * Possibly related: "to set the context for the collaborative reasoning capabilities for frontier models;" what does stronger reasoning models "setting the context" mean?
* Are all the models used for self-play training (A, B, and the answer extraction model) the same? Is a different model (maybe stronger) used to extract answers?
* How does LLM-as-a-judge for answer extraction work? The appendix acknowledges that the extraction is not perfect. Does the extract model faithfully summarize each "agent's" reasoning? Does it confuse the agents? Is it able to correctly identify when no clear answer has been arrived at for one agent (and possibly only one of the two agents)?

**Ethical Concerns:**

["NO or VERY MINOR ethics concerns only"]

**Final Justification:**

Given the effectiveness exhibited in cross-model collaboration settings, I have revised my assessment from 3 to 4.

The method, experiments, and results are sound, and I don't see grounds for rejection based on those alone. The presentation as a way to train models for collaborative task solving raises questions about the types of collaborative approaches this paper experiments with, and here I'm concerned about how different these settings are from single-agent task solving (since both agents have full observation over the other agent's reasoning and actions, and act in the same way here). The authors acknowledge that this work illustrates the potential to extend to collaborative, partially observable (or other collaborative settings with non-identical observation/action) but experiments for that are (understandably) not possible in the time frame of the author response. This motivates my borderline rating.

**Limitations:**

yes

**Quality:**

3

**Strengths And Weaknesses:**

Strengths:
* New multi-agent task with concrete functional definitions of notions of agreeableness and persuasiveness
* Useful software tool for simulating interactions between multiple systems

Weaknesses:
* I may be missing something here, but it is not clear to me how these experiments evaluate collaboration between multiple distinct agents. If there are two agents A and B which both use the same underlying model generating a sequence of responses $(a_1, b_1, a_2, b_2, \ldots)$, it is almost identical to the setup where A generates a sequence of responses $(a_1, a_2, a_3, a_4 \ldots)$, since $b_1 = LLM_B(a_1) = LLM_A(a_1) = a_2$ and so on. Since the prompt also appears to be the same for both A and B, the only difference would be the sequence of user/assistant tags in the serialized messages. This is somewhat like self-refine, with the constraint of the two "agents" having to agree resembling self-consistency. This combination of the two methods could be a novel contribution in its own right, but it should be framed and analyzed as such, with relevant experiments to show differences from the base methods as single-agent reasoning approaches. However, the experiments in this paper don't really address how this evaluates collaboration.
  * There are experiments where the training method is shown to generalize to other agents, but the only ones evaluated are versions of the same model without training. While these experiments are informative, really evaluating the collaborative abilities of models would have to go much further beyond given the broad scope adopted in the introduction (lines 30–32).
  * The experiments do show improvements (especially with the proposed self-play training approach), but it's unclear whether this is fundamentally due to a difference between multiple distinct agents, or whether there is an explanation in terms of a single agent (more steps in CoT – standard CoT involves one step while the consistency condition here has the reasoning proceed for more steps – or better responses to mistakes in contexts, etc.)

---

> ### Author Rebuttal · Authors · 2025-07-31
>
> While we are happy to hear that you find the concrete definitions of social metrics and our software infra useful, we apologize if some of the motivations are unclear or certain experiment results are missing to justify the claims.
>
> We strive to respond to your questions and concerns here:
> 1. **Same underlying model for collaboration.** Sorry for the confusion, we would like to clarify that the reason we mainly use the same model to collaborate (i.e., self-collaboration setting) is in two folds:
>   * a. *Creating single-source, distillation-free training data* To create the training data, we opt for self-play using the same model as the model we train, so all the data comes from a single-source/model. In this way, we avoid the confounder that a different model brings in the training process.
>   * b. *Construct fair comparison with single-agent methods* Albeit a minor reason, the self-collaboration setting during evaluation also allows us to have a direct comparison with single-agent methods since no other models are used.
>
> However, upon reflection with your feedback, we do agree that more cross-model evaluations are needed to properly support the broader scope we introduced. Thus we conducted experiments to compare the Llama and Coral models when collaborating with a set of different collaborators from a variety of model families (agent A starts the conversation with the question):
>
> > **[Cross-model collaboration comparison on MMLU-Pro]**
> | ↓ Agent B / Agent A → | Qwen2.5-7B-Instruct | GPT-4o | Gemini-1.5-Pro | Claude-3.7-Sonnet | Avg. Perf. |
> |-----------------|---------------------|--------|----------------|-------------------|------------|
> | Llama-8B        | 52.9                | 69.7   | 68.7           | 79.3              | 67.7       |
> | Coral-8B        | 65.1 (+12.2)        | 71.2 (+1.5) | 73.2 (+4.5)  | 80.8 (+1.5)      | 72.6 (+4.9)|
> | Llama-70B       | 58.1                | 71.7   | 73.2           | 76.6              | 69.9       |
> | Coral-70B       | 72.2 (+14.1)        | 77.7 (+6.0) | 75.1 (+1.9)  | 82.5 (+5.9)      | 76.9 (+7.0)|
>
> > **[Cross-model collaboration comparison on ExploreToM]**
> | ↓ Agent B / Agent A → | Qwen2.5-7B-Instruct | GPT-4o | Gemini-1.5-Pro | Claude-3.7-Sonnet | Avg. Perf. |
> |-------------------|---------------------|--------|----------------|-------------------|------------|
> | Llama-8B          | 43.9                | 47.6   | 50.6           | 44.2              | 46.6       |
> | Coral-8B          | 82.6 (+38.7)        | 79.6 (+32.0) | 71.9 (+21.3)  | 81.8 (+37.6)      | 78.8 (+32.2)|
> | Llama-70B         | 64.1                | 72.6   | 69.6           | 82.5              | 72.2       |
> | Coral-70B         | 93.1 (+29.0)        | 89.1 (+16.5) | 84.5 (+14.9)  | 89.2 (+6.7)      | 89.0 (+6.8)|
>
> From these additional results, we can see that Coral models, which are trained with the self-play data, generalizes well when paired with a diversified set of collaborators.
>
> 2. **Improvements due to single-agent ability or collaboration.** This is a great question. While it is arguably hard to completely separate the collaboration and reasoning capabilities, we attempt this by evaluating the CoT capabilities of the Coral models trained for collaborative reasoning, and show the results on MMLU-Pro as follows:
>
> > **[Evaluation of Coral-trained models on MMLU-Pro]**
> | Model                  | CoT Evaluation | Coral Evaluation   |
> |------------------------|----------------|--------------------|
> | Llama-3.1-8B-Instruct  | 44.4           | 45.6               |
> | Coral-8B               | 46.7 (+2.3)    | 59.8 (+14.2)       |
> | Llama-3.1-70B-Instruct | 63.8           | 65.8               |
> | Coral-70B              | 67.2 (+3.4)    | 75.2 (+9.4)        |
>
> From this table, we can see that the self-play-based collaborative training does also improve CoT reasoning capabilities, albeit yielding less boost than in the collaborative setting. Based on these results + the human evaluation, we conjecture that the training improves both reasoning and collaboration skills.
>
> 3. **Source of the training data.** We use the questions from the training split of the corresponding dataset to be the source of the training data to generate the conversations (as noted in D.1 section in the appendix), and out-of-distribution evaluation results can be found in Tab. 3. We will make this clearer in the main body of the paper in the next version.
>
> 4. **stronger reasoning models "setting the context".** Sorry for the confusion caused by the wording here, we simply mean the performance of these stronger reasoning models (e.g., O1, Gemini-2.5-flash) are not baselines we aim to beat, but to provide more reference points for how well our models are doing.
>
> 5. **Details for the extractor.** As for the extractor, for the same reasons we mentioned in 1.a and 1.b above, we use the same model to perform the belief extractions (e.g., Llama-8B as extractor for Llama-8B collaborations). And for the input of the extractor, we only consider the *current turn* instead of the whole chat history, this means the extractor is simply trying to *extract* rather than *summarize*, as we can also see its instruction Tab. 8 in the appendix. As for the extraction accuracy when the extractor outputs “not sure”, we found that it is mostly because the response is half-way towards a final answer, or that it is asking questions (e.g., for confirmation). We will add these details of the extractors in the updated version.
>
> Thank you again for your questions and constructive feedback. We hope the additional discussions and results above could (at least partly) resolve your concerns, and feel free to follow up if you have any more questions!

---

> > ### Comment · Reviewer_ygtt · 2025-08-04
> > **CoT vs Coral**
> >
> > The question about CoT vs Coral wasn't about whether Coral improved CoT reasoning (which the presented results show), but whether there was something about it specifically being collaborative that improved reasoning. For example, what if you just asked the single model that is using CoT to attempt the problem again given both the question and the previous CoT? This would be a second attempt at the problem while having access to previous reasoning. The Coral without training results from Figure 3 suggest that this does about as well as CoT. One way to see these results is that it is the single agent reasoning ability (both starting with an empty slate, and conditioning on the previous reasoning) that matters, and not the fact that this is a collaborative system.

---

> > > ### Author Response · Authors · 2025-08-05
> > > **RE: CoT vs Coral**
> > >
> > > Thanks for your recognition for the new results and we apologize for any confusion regarding CoT vs. Coral.
> > >
> > > 1. **Single-agent reasoning or collaborative system.** We would like to emphasize that our main goal of the paper is to advance social agents by training the LLMs to better collaborate with others on reasoning tasks. Such improvements can come from better skills in identifying and fixing a mistake, which are certainly applicable through the lens of a single-agent used in a self-refine / self-reflect setting. However, there are other skills developed through collaboration such as more effective disagreement and more natural conversation, for which we show through human annotations as Fig. 5.
> > > 2. **Results in Fig. 3.** The self-collaborative Coral setting shown in Fig. 3 is indeed similar to self-refine, and more of such results are actually presented in Tab. 1 and discussed in Section 3 as part of our main motivation – before the collaborative training, the model can not consistently use such multi-turn collaboration to improve its performance over CoT. Regarding extra experiments on evaluating or training for self-refine, we regret that we are not able to produce those given the short time window.
> > > 3. **More discussions with self-refine methods.** As discussed above and in the related work section, while self-refinement (e.g., multi-round refinement conditioned on previous reasoning) is similar in format as it also uses multi-turns to improve reasoning. However, its motivation is purely from improving reasoning and not advancing collaboration, which is the main goal of this work. This means that it can not be applied to tasks that are natively multi-agent, such as negotiation or partially-observable collaborative games (e.g., Hanabi), which we are actively pursuing.
> > > 4. **Single-agent CoT refinement and multi-agent Coral are orthogonal.** Lastly, one could also argue that any single-agent refinement methods can be used in combination with multi-agent collaboration, as each agent can self-reflect to improve individual responses, and then the collaboration will push the quality even higher. And we leave this as important future work.

---

> > > > ### Comment · Reviewer_ygtt · 2025-08-06
> > > >
> > > > I've increased my rating for the paper based on he additional cross-model collaboration results presented.
> > > >
> > > > > This means that it can not be applied to tasks that are natively multi-agent, such as negotiation or partially-observable collaborative games (e.g., Hanabi), which we are actively pursuing.
> > > >
> > > > In these "natively multi-agent" setting, the constraints of the task (partial observability, etc.) strictly ensure that agents are different and a self-refine analogy is not possible. However, in the reasoning benchmarks presented in this paper, there is no such inherent restriction, and any restriction is based on system design. While I agree that the proposed method _could_ generalize to such situations, the experiments presented don't show this, so we don't know. What is presented is a setting where both agents have identical observation (the prompt for each agent is almost the same, with some minor changes in user/assistant tags), so it is hard to clearly delineate it from a "training a model to engage in self-refinement" experiment.
> > > >
> > > > > Regarding extra experiments on evaluating or training for self-refine, we regret that we are not able to produce those given the short time window.
> > > >
> > > > This is understandable. The cross-model results partly (but not conclusively) address my concern, and I have changed by assessment to reflect that.

---

> > > > > ### Author Response · Authors · 2025-08-08
> > > > > **Thank you**
> > > > >
> > > > > We are glad to hear that the cross-model evaluation results partly addressed the concern, and we would like to thank you for the kind consideration and the update of the rating.
> > > > >
> > > > > Regarding "natively multi-agent" settings, we definitely understand that we did not experiment for these settings in the scope of this work, but as it is something we are actively working on as "future" work, we hoped that it could help understand our motivation and rationalize some of the design choices we made for this work. As you said, we also believe our proposed methods and infra *could* generalize to these natively multi-agent scenarios, which we are pursuing at the moment.
> > > > >
> > > > > Thank you again for the engaging discussion!

---

### Official Review · Reviewer_Ecdm · 2025-07-01

**Clarity:** 4
**Significance:** 4
**Originality:** 4
**Rating:** 6
**Confidence:** 4

**Summary:**

This paper focuses on evaluating if LLMs are able to collaborate with each other on various reasoning tasks: math, coding, etc. The authors propose an evaluation framework Coral whose main metric is "agreement correctness" -> Do the two models working together agree with each other and are they right.

The authors find LLMs aren't the best at collaborating and in order to improve this built a data generator named Matrix that generates synthetic conversations to finetune LLMs for collaboration and find this improves performance.

The contributions as I see it are:

1) The synthetic dataset created to improve collaboration capabilities amongst LLMs.

2) The Matrix framework which claims to be able to generate a large number of conversations across multiple LLMs in a scalable manner.

**Questions:**

Questions

1) Did you try pairing up the best collaborators? Such as Claude 3.7 / Gemini 2.5 flash. As far as I can tell all the combinations consist of the same base model.

**Ethical Concerns:**

["NO or VERY MINOR ethics concerns only"]

**Final Justification:**

My score was already a Strong Accept and the authors answered any small questions I had and ran additional experiments that I thought were important and interesting.

**Limitations:**

yes

**Quality:**

4

**Strengths And Weaknesses:**

Strengths

1) The topic of this paper is very interesting and I think important for the community to explore. It's also well-written and generally clear to follow.

2) Both the synthetic dataset and the Matrix framework seem useful for the community.

3) There are strong baselines the authors compare against to prove the usefulness of their dataset

Weaknesses

1) One detail that I didn't see (unless I missed it) was the number of conversations being generated and finetuned on.

2) In addition to self-model collaboration I think cross-model collaboration would be important to put. I don't mean the Coral-trained model with its untrained counterpart (which was done and great to see) but I mean something like Claude collaborating with Gemini (which were the best base models).

---

> ### Author Rebuttal · Authors · 2025-07-31
>
> We would like to thank the reviewer Ecdm for your strong support of this paper and we are glad to hear you find the topic very interesting and you recognize the value of the synthetic dataset and matrix framework to the community.
>
> To respond to your questions and concerns:
> 1. **Details on the size of the synthetic data.** Thanks for bringing this up! For each example in the training split of the dataset, we sample 5 trees with the width (i.e., expansion size) of 5 for DPO. These hyperparameters are also noted in appendix D.2 Line 969-970, and the sizes of the original training split of each dataset is also noted in appendix D.1. However, we do realize we did not share the size of the turn-based synthetic data **after filtering**, and per your suggestion, we calculated and summarized such statistics below:
> > **[Number of turns for DPO training]**
> | Generation Model         | MBPP-CR | MATH  | MMLU-Pro | ExploreToM |
> |--------------------------|---------|-------|----------|------------|
> | Llama-3.1-8B-Instruct    | 33.8K   | 85.1K | 160.6K   | 100.1K     |
> | Llama-3.1-70B-Instruct   | 33.3K   | 88.5K | 99.8K    | 89.7K      |
>
> We will include this table as well as more statistics on the training data in the next version of the paper.
> 2. **Cross-model collaboration.**  Thanks for your suggestion, we agree that a wider-range of cross-model collaboration results would help us understand the generalizability of the Coral models to collaborate with other models beyond its own model family. We are also intrigued by this research question, thus we conducted the following experiments to pair trained (Coral) and untrained (original) Llama models (as Agent B) with *a committee of diverse collaborators* as agent A (who starts the conversation with the question):
>
> > **[Cross-model collaboration comparison on MMLU-Pro]**
> | ↓ Agent B / Agent A → | Qwen2.5-7B-Instruct | GPT-4o | Gemini-1.5-Pro | Claude-3.7-Sonnet | Avg. Perf. |
> |-----------------|---------------------|--------|----------------|-------------------|------------|
> | Llama-8B        | 52.9                | 69.7   | 68.7           | 79.3              | 67.7       |
> | Coral-8B        | 65.1 (+12.2)        | 71.2 (+1.5) | 73.2 (+4.5)  | 80.8 (+1.5)      | 72.6 (+4.9)|
> | Llama-70B       | 58.1                | 71.7   | 73.2           | 76.6              | 69.9       |
> | Coral-70B       | 72.2 (+14.1)        | 77.7 (+6.0) | 75.1 (+1.9)  | 82.5 (+5.9)      | 76.9 (+7.0)|
>
> > **[Cross-model collaboration comparison on ExploreToM]**
> | ↓ Agent B / Agent A → | Qwen2.5-7B-Instruct | GPT-4o | Gemini-1.5-Pro | Claude-3.7-Sonnet | Avg. Perf. |
> |-------------------|---------------------|--------|----------------|-------------------|------------|
> | Llama-8B          | 43.9                | 47.6   | 50.6           | 44.2              | 46.6       |
> | Coral-8B          | 82.6 (+38.7)        | 79.6 (+32.0) | 71.9 (+21.3)  | 81.8 (+37.6)      | 78.8 (+32.2)|
> | Llama-70B         | 64.1                | 72.6   | 69.6           | 82.5              | 72.2       |
> | Coral-70B         | 93.1 (+29.0)        | 89.1 (+16.5) | 84.5 (+14.9)  | 89.2 (+6.7)      | 89.0 (+6.8)|
>
> From these results, we can see that through collaborative training from self-play, the models not only collaborate better with themselves or untrained models in the same model family, but it generalizes quite well to collaborators from a variety of model families as well.
>
> We will definitely include these results in the next version of the paper.
>
> 3. **Pairing up best collaborators.** That is an interesting question that we are intrigued to find out as well, so we compiled the following experiment results on MMLU-Pro to pair claude-3.7-sonnet with Gemini-2.5-flash and O1, which are the strongest reasoning models we have access to:
> | Agent B       | Agent A      | Agreement Correctness |
> |---------------------|---------------------|-----------------------|
> | Claude-3.7-Sonnet   | Claude-3.7-Sonnet   | 79.1                  |
> | O1                  | O1                  | 82.8                  |
> | Gemini-2.5-flash    | Gemini-2.5-flash    | 81.0                  |
> | Claude-3.7-Sonnet   | O1                  | 85.0                  |
> | Claude-3.7-Sonnet   | Gemini-2.5-flash    | 82.7                  |
>
> Here we can see that pairing strong models together can actually go beyond the max performance of the self-collaboration settings, hinting exciting future research in this domain.
>
> We thank the reviewer again for your suggestions and strong support of our work, as we will try to incorporate the discussions and new results above in the next version of our paper!

---

> > ### Comment · Reviewer_Ecdm · 2025-08-06
> >
> > Thank you for the detailed information. The results on pairing up best collaborators is very interesting.
> >
> > Overall, all my questions for have been resolved.

---

### Official Review · Reviewer_MWLw · 2025-07-02

**Clarity:** 3
**Significance:** 3
**Originality:** 3
**Rating:** 5
**Confidence:** 4

**Summary:**

This paper introduces a new collaborative reasoning approach that uses LLMs to self-chat when solving problems. The authors first investigate the behavior of current LLMs using this approach out-of-the-box and found that it did not consistently outperform chain-of-thought, potentially due to a lack of effective disagreement. They then propose a self-training approach involving tree-sampling, belief-filtering, and preference fine-tuning to synthesize data, filter data, and train models without human annotation to improve performance on this collaborative reasoning approach. Results show that after this training, the model demonstrates stronger performance than direct CoT SFT training on most test benchmarks.

**Questions:**

Please refer to the weakness part:
1. The model trained with this self-training approach appears to be specifically optimized for collaborative reasoning. What is their performance when used in single-agent scenarios, which are more widely used for solving different tasks? Will this specific format training make it perform worse without the collaborative reasoning approach?
2. Although it is compared with CoT+SFT and CoT+DPO and shows better performance, the computational cost appears much higher. It is necessary to provide cost-performance analysis to gauge the trade-off.
3. When measuring the belief state for each turn, there might exist intermediate steps/turns where the agent has not reached a final answer. How is the belief extracted in such cases?
4. Additionally, is social behavior (persuasive and assertive) taken into consideration when filtering the data for model training, as the authors suggested this might be a reason that hinders effective utilization of this collaborative reasoning approach?

**Ethical Concerns:**

["NO or VERY MINOR ethics concerns only"]

**Final Justification:**

During rebuttal, most of my concerns have been explained by the authors. Therefore, I am inclined to accept.

**Limitations:**

The authors have discussed several limitations of this work, including robustness of belief extraction and agreement measures. However, the limitation that the trained model is specifically designed for this collaborative reasoner approach and might not work well when used in single-agent scenarios should be explored and discussed.

**Quality:**

3

**Strengths And Weaknesses:**

### Strengths
1. This paper introduces a new and interesting way to solve reasoning tasks with LLMs by letting them self-chat to solve tasks collaboratively.
2. Preliminary experiments are conducted to identify the lack of effective disagreement capability that hinders effective utilization of this approach. An effective self-training approach is then proposed to synthesize data, filter data, and train models without human annotation to improve performance on this collaborative reasoning approach.
3. The proposed self-training approach is compared with different training methods like CoT+SFT and CoT+DPO, and various LLMs are evaluated. Results show that the self-training approach is more effective and also demonstrate generalization capability.
4. A versatile backend for multi-agent inference is proposed, which supports models with different backends and enables fast inference.

### Weaknesses
1. The model trained with this self-training approach appears to be specifically optimized for collaborative reasoning. What is their performance when used in single-agent scenarios, which are more widely used for solving different tasks? Will this specific format training make it perform worse without the collaborative reasoning approach? This should be explored and discussed.
2. The self-training approach incurs high costs for data synthesis and filtering. Although it is compared with CoT+SFT and CoT+DPO and shows better performance, the computational cost appears much higher. It is necessary to provide cost-performance analysis to gauge the trade-off.
3. When measuring the belief state for each turn, there might exist intermediate steps/turns where the agent has not reached a final answer. How is the belief extracted in such cases? Additionally, is social behavior (persuasive and assertive) taken into consideration when filtering the data for model training, as the authors suggested this might be a reason that hinders effective utilization of this collaborative reasoning approach?

---

> ### Author Rebuttal · Authors · 2025-07-31
>
> We would like to thank the reviewer MWLw for the careful and positive review of our work, and we are glad to know that you find the self-chat approach interesting and effective.
>
> Here we respond to the questions and concerns:
> 1. **Performance of collaborative models used in single-agent scenarios.** Thanks for the suggestion! While improving collaborative reasoning skills is the main focus of this work, we agree that it is beneficial to understand how our coral model performs under usual single-agent setup for even broader use cases. Following your suggestion, here we report:
>
> > **[Single-agent CoT performance on our trained Coral models]**
> | Model                  | MMLU-Pro | ExploreToM |
> |------------------------|----------|------------|
> | Llama-3.1-8B-Instruct  | 44.4     | 60.8       |
> | Coral-8B               | 46.7 (+2.3) | 91.9 (+31.1) |
> | Llama-3.1-70B-Instruct | 63.8     | 71.3       |
> | Coral-70B              | 67.2 (+3.4) | 93.5 (+32.2) |
>
> From these results, we can see that our collaborative self-training not only didn’t hurt the single-agent CoT performance, but actually improved it. Thanks for pointing us to these experiments and we would also incorporate them into our next version of the paper.
>
> 2. **Computational cost.** Thanks for bringing this up! We would first like to clarify that while we report single-agent CoT + DPO/SFT results to add more context for the Coral results, our main focus is still improving models’ collaboration skills measured by Coral performance. And it is not our intention to argue that Coral is better than CoT for reasoning despite the results showing so, as they have distinct use cases. Regarding compute cost, we calculated the average prompt / response length for Coral and CoT training data, and it is listed as follows:
> | Settings          | Prompt (# tokens) | Response (# tokens) |
> |-------------------|-------------------|----------------------|
> | Single-Agent CoT  | 289.0             | 372.2                |
> | Coral             | 533.3             | 318.9                |
>
> While the prompt length under Coral setting is 89% longer than the CoT setting, the response length is 15% shorter. Since no loss is computed on the prompt, we estimate that the training cost between the two are comparable as well. We will add such discussion about compute cost trade-off in the next version of the paper.
>
> 3. **Belief extraction without final answer.** Our belief extraction method actually considers this exact challenge (section 2.2 Line 104-106) as we instruct the extractor to say “not sure” in this case (section 2.2 Line 116), among other challenges we discussed for belief extraction in the beginning of section 2.2. We will try to make this a bit clearer in the next version of the paper.
> 4. **Filtering based on social behavior.** That’s a great question as we can certainly filter the data based on the social metrics. However, we’d like to emphasize that the purpose of social metrics is for observing and analyzing the behavior of the models under different scenarios (e.g., before/after training, different tasks, pairing different models, etc), so that they provide different perspectives than reasoning correctness. Thus we refrain ourselves from constructing training examples based on such social metrics so we can observe the change of behaviors in a neutral way.
>
> Thanks again for your questions and we hope the additional results and discussions (which we will also incorporate in the next version of the paper) resolves at least parts of your concerns. Please do not hesitate to let us know if you have any further questions that can convince you to rate our work in a more favorable way!

---

> > ### Comment · Reviewer_MWLw · 2025-08-05
> > **Follow-up questions**
> >
> > Thank you for your explanations, which have clarified some of my concerns. I also have a question about the behavior of Coral models in single-agent scenarios. Do they exhibit self-refine, self-chat behaviors, within their CoT? Furthermore, what is their CoT length, and is it longer than what the CoT models?

---

> ### Author Response · Authors · 2025-08-08
> **RE: Follow-up questions**
>
> Thanks for following up!
>
> We did a quick check for `Llama-3.1-70B-Instruct` and its trained versions on the MMLU-Pro dataset. Regarding CoT length, both the trained Coral model and CoT model actually generate much longer reasoning chains when evaluated in a CoT setting: Coral-trained (677.6 tokens) vs. CoT-trained (644.6 tokens) vs. original Llama (426.8 tokens). Regarding any behavior change, we did a qualitative analysis by manually looking into around 20 examples. And we found that the coral-trained model do exhibit more self-refine behavior even during CoT evaluation, such as saying *"However, I should also check ..."*. These findings are aligned with our previous observation that the CoT performance is not only preserved by the coral-trained models, but actually improved, shown not only by the performance numbers but also the length of the reasoning chains as well as reasoning behaviors.
>
> We would like to thank you again for sparking these interesting discussions and pointing us to the new findings, as we will try to incorporate them in the camera-ready version (if the paper is accepted).

---

### Official Review · Reviewer_5zuY · 2025-07-08

**Clarity:** 3
**Significance:** 2
**Originality:** 2
**Rating:** 4
**Confidence:** 2

**Summary:**

This paper introduces Collaborative Reasoner (Coral), a framework designed to evaluate and improve the collaborative reasoning abilities of LLMs in multi-agent, multi-turn conversational settings. The authors propose a training strategy using self-play to generate large-scale synthetic multi-turn conversations. These are then filtered and used for preference-based fine-tuning using DPO. The framework also introduces Matrix, a high-throughput system for scalable synthetic data generation. Experimental results across different tasks (math, coding, scientific QA, social reasoning, etc.) show that collaborative fine-tuning improves agreement correctness substantially over CoT baselines.

**Questions:**

1. The paper introduces multiple components, including multi-turn collaborative reasoning, synthetic conversation data, and the Matrix infrastructure. Could the authors clarify the conceptual centerpiece of the paper? What is the single most important innovation they aim to establish?

2. The social behavior metrics (e.g., persuasiveness, assertiveness) are interesting but may be considered subjective. Could the authors elaborate on the practical need for these specific metrics? What are some concrete application scenarios where these metrics make a tangible difference?

3. Given the reliance on synthetic data, how well do the trained models generalize to real-world collaborative settings?

**Ethical Concerns:**

["NO or VERY MINOR ethics concerns only"]

**Final Justification:**

While I appreciate the authors’ response, it did not provide new, concrete information that would help me reassess my evaluation. It’s possible that I may not have fully understood certain aspects, so I am choosing to lower my confidence accordingly.

**Limitations:**

Yes.

**Paper Formatting Concerns:**

No major formatting issues found.

**Quality:**

3

**Strengths And Weaknesses:**

**Strengths:**

1. The paper is well-structured and clearly written, making it easy to follow the motivation, methodology, and findings. Illustrative examples and figures (e.g., Fig. 1) are particularly helpful for understanding the proposed setting.

2. The authors provide solid experimental details, code, and metrics across diverse reasoning domains (math, coding, QA, social reasoning). This level of transparency strengthens reproducibility and trust in the findings.

3. The proposed Coral framework and the Matrix infrastructure are valuable in practice and could serve as foundational tools for future work on social LLM agents and collaborative multi-agent systems.

**Weaknesses:**

1. While the paper integrates several components (multi-turn conversations, synthetic data generation, preference tuning, scalable infrastructure), the contribution may appear as an aggregation of known ideas rather than a fundamentally novel concept. The core innovation could be better distilled and emphasized.

2. The proposed social behavior metrics, such as persuasiveness and assertiveness, though intuitive, raise questions about their objectivity and necessity. Their practical utility remains somewhat vague, especially without strong connections to downstream applications or human-agent collaboration scenarios.

3. The paper could benefit from a clearer explanation of what constitutes a "fair" comparison for the proposed framework of social agents.

---

> ### Author Rebuttal · Authors · 2025-07-31
>
> We would like to thank the reviewer 5zuY for the thoughtful review and the support of our work. We are glad to hear that you find our paper clearly written with solid experiments details, and our proposed framework and infrastructure valuable.
>
> To respond to your questions and concerns regarding this work:
> 1. **Conceptual centerpiece.** We apologize that the innovations seem a bit scattered and we definitely agree that we didn’t invent all the techniques (e.g., preference tuning w/ DPO). We would like to emphasize that as the first work (to the best of our knowledge) to advance collaborative skills of social agents, the coral framework itself is the centerpiece that ties all the innovations (i.e., agreement-based evaluation, social metrics, conversational self-play, turn-based preference turning) together. Through the coral framework, we show that it is possible to train the LLMs to be better at robust collaborative reasoning that generalizes beyond the model families as well as OOD tasks. And while many evaluation and modeling choices seem natural post hoc, finding the correct recipe and building an accompanying infra (i.e., Matrix) takes a non-trivial amount of exploration. For example, we were stuck with the underwhelming results of SFT before adopting DPO to allow finer-grain preference tuning given the same conversation prefix.
> 2. **Practical utility of social metrics.** We would like to clarify that the purpose of such social metrics is to provide lens into understanding the behaviors of the models in different scenarios (e.g., before/after training, different tasks, pairing different models, etc), and *not* something we seek to directly optimize. Thus we deliberately disentangle such social behaviors from reasoning correctness to provide different perspectives. Moreover, while the metrics themselves are based on subjective terms, how we implement them (i.e., by comparing change of beliefs) is in a purely objective manner, thus it should be a fair comparison to make observations across all evaluations. Lastly, there are various practical use-cases where such metrics may be useful in understanding and improving human-AI interactions such as persuasion for social good [Wang et al. 2019], reducing polarization [Bai et al. 2025] etc. We will add these references to emphasize the value of the social metrics.
> 3. **Synthetic data → real-world collaborations.** Thanks for bringing this up. We believe that “synthetic” is not necessarily opposite to being “real-world”, as we show that our methodology can be applied to a wide range of tasks, and the trained models can be generalized to other tasks in a similar domain (Tab.3). In addition, the human evaluation results (Fig. 5) also suggest that our coral models are on the path to improved conversational and social capabilities, which are crucial for real-world social interactions.
> 4. **Fair comparison of social agents.** In this work, we propose performance-related (i.e., agreement correctness) metrics which are our main goals for improvements. In addition we propose the social metrics to observe the behaviors of the social agents and not directly optimize the models towards them. We believe all the comparison between the social agents under the collaborative reasoning settings are fair comparisons, but would be happy to answer more questions if you would like more detailed information.
>
> Hopefully these responses would help answer your questions, and we would also incorporate these discussions into the next version of our paper. Please do not hesitate to follow up if you have further questions!
>
> **References**
> [Wang et al., 2019] *Wang, Xuewei, et al. "Persuasion for Good: Towards a Personalized Persuasive Dialogue System for Social Good." Proceedings of the 57th Annual Meeting of the Association for Computational Linguistics. 2019.*
> [Bai et al., 2025] *Bai, Hui, et al. "LLM-generated messages can persuade humans on policy issues." Nature Communications 16.1 (2025): 6037.*

---

> > ### Comment · Reviewer_5zuY · 2025-08-05
> >
> > Thank you for the rebuttal. I appreciate the clarifications on the overall framing and the intended role of the social metrics.

---

### Note · Authors · 2025-08-15

We would like to thank all the reviewers for their careful initial reviews and the engaging discussions during the rebuttal phase.

We are happy to see that most of the initial reviews are leaning towards acceptance and we are even more glad to see that the additional discussions and results during the rebuttal phase were appreciated by the reviewers in helping resolve their concerns. In particular, reviewer Ecdm mentioned that “*all the questions have been resolved*” and reviewer ygtt decided to “*increase my rating for the paper based on the additional cross-model collaboration results presented*”.

As the first work focusing on advancing collaborative skills of social agents, we really appreciate the support as well as the constructive feedback from the reviewers, and we will definitely incorporate the new discussions and results in the final version of the paper.

---

### Decision · Program_Chairs · 2025-09-17

**Decision:**

Accept (poster)

**Comment:**

This paper studies whether LLMs can collaborate effectively in reasoning tasks (math, coding, QA, social reasoning) and finds that existing models have weak collaborative reasoning skills due to overly agreeable or unproductive conversational behaviors. The lack of effective disagreement capability makes it difficult to leverage a collaborative reasoning framework, which (with existing models) offers no benefit over single-agent reasoning. This is an important finding that highlights a gap between what LLMs can do in isolation and what is required for real-world multi-agent or human-AI collaboration.

On the positive side, the paper contributes the Coral (Collaborative Reasoner) framework, which evaluates collaborative reasoning through multi-turn tasks requiring agents to disagree with incorrect answers, persuade each other, and ultimately reach a correct joint solution. Based on this framework, the authors introduce a self-play training method that generates synthetic multi-turn conversations to improve multi-turn collaborative reasoning skills. Experiments show that this approach substantially improves agreement correctness over chain-of-thought baselines, generalizes across model families, and even enhances single-agent performance. The paper also offers human evaluations that show more natural conversational behavior.

On the negative side, it is somewhat difficult to tease apart the benefit that comes from single-agent refinement and the benefit of true collaboration. Secondly, some of the social metrics such as persuasiveness and assertiveness seem rather subjective, and it was not clear to reviewers if and why they are needed. Another issue was that the computational cost and efficiency trade-offs were initially underexplored.

In the rebuttal and discussion, the authors provided various clarifications and new results, such as compute trade-off analysis showing comparable costs to CoT. The overall reviewer sentiment is that the work should be accepted, with reviewers noting that “all questions have been resolved,” and the consensus leaning toward acceptance.

Overall, I also think this paper should be accepted. It provides the first comprehensive framework for evaluating and improving LLM collaborative reasoning, with both practical infrastructure (Matrix) and conceptual advances (agreement-based evaluation, synthetic self-play training).